# Evolution of asymmetric gamete signaling and suppressed recombination at the mating type locus

Zena Hadjivasiliou[1,2,3]*, Andrew Pomiankowski[2,3]

[1]Department of Biochemistry, University of Geneva, Geneva, Switzerland; [2]Centre for Mathematics and Physics in the Life Sciences and Experimental Biology (CoMPLEX), University College London, London, United Kingdom; [3]Department of Genetics, Evolution and Environment, University College London, London, United Kingdom

**Abstract** The two partners required for sexual reproduction are rarely the same. This pattern extends to species which lack sexual dimorphism yet possess self-incompatible gametes determined at mating-type regions of suppressed recombination, likely precursors of sex chromosomes. Here we investigate the role of cellular signaling in the evolution of mating-types. We develop a model of ligand-receptor dynamics, and identify factors that determine the capacity of cells to send and receive signals. The model specifies conditions favoring the evolution of gametes producing ligand and receptor asymmetrically and shows how these are affected by recombination. When the recombination rate evolves, the conditions favoring asymmetric signaling also favor tight linkage of ligand and receptor loci in distinct linkage groups. These results suggest that selection for asymmetric gamete signaling could be the first step in the evolution of non-recombinant mating-type loci, paving the road for the evolution of anisogamy and sexes.
DOI: https://doi.org/10.7554/eLife.48239.001

*For correspondence:
zena.hadjivasiliou@unige.ch

Competing interests: The authors declare that no competing interests exist.

## Introduction

Sex requires the fusion of two cells. With few exceptions, the sexual process is asymmetric with partnering cells exhibiting genetic, physiological or behavioral differences. The origins of sexual asymmetry in eukaryotes trace back to unicellular organisms with isogametes lacking any size or mobility difference in the fusing cells (*Parker et al., 1972*; *Bell, 1978*; *Charlesworth, 1978*; *Hoekstra, 1987*; *Randerson and Hurst, 2001*; *Lehtonen et al., 2016*). Isogamous organisms are divided into genetically distinct mating types, determined by several mating type specific genes that reside in regions of suppressed recombination (*Fraser et al., 2004*; *Ahmed et al., 2014*; *Branco et al., 2017*; *Branco et al., 2018*). The morphologically identical gametes mate disassortatively, scarcely ever with members of the same mating type. It follows that only individuals of a different mating type are eligible mating partners. This arrangement poses a paradox as it restricts the pool of potential partners to those of a different mating type, introducing a major cost (*Hoekstra, 1987*).

Several hypotheses have been proposed to explain the evolution of isogamous mating types (*Billiard et al., 2011*; *Billiard et al., 2012*; *Perrin, 2012*). Mating types could serve as a restrictive mechanism preventing matings between related individuals thereby avoiding the deleterious consequences of inbreeding (*Charlesworth and Charlesworth, 1979*; *Uyenoyama, 1988*; *Czárán and Hoekstra, 2004*). Another idea is that mating types facilitate uniparental inheritance of mitochondria, which leads to improved contribution of the mitochondrial genome to cell fitness (*Hastings, 1992*; *Hurst and Hamilton, 1992*; *Hutson and Law, 1993*; *Hurst, 1996*; *Hadjivasiliou et al., 2012*; *Hadjivasiliou et al., 2013*; *Christie et al., 2015*; *Christie and Beekman, 2017a*). Both

**eLife digest** Sexual reproduction, from birds to bees, relies on distinct classes of sex cells, known as gametes, fusing together. Most single cell organisms, rather than producing eggs and sperm, have similar sized gametes that fall into distinct 'mating types'. However, only sex cells belonging to different mating types can fuse together and sexually reproduce.

At first glance, it seems illogical that cells from the same mating type cannot reproduce with each other, as this restricts eligible partners within a population and makes finding a mate more difficult. Yet the typical pattern amongst single cell organisms is still two distinct classes of sex cells, just as in birds and bees. How did the obsession with mating between two different types become favored during evolution?

One possibility is that cells with different mating types can recognize and communicate with each other more easily. Cells communicate by releasing proteins known as ligands, which bind to specific receptors found on the cell's surface. Using mathematical modelling, Hadjivasiliou and Pomiankowski showed that natural selection typically favors 'asymmetric' signaling, whereby cells evolve to either produce receptor A with ligand B, or have the reverse pattern and produce receptor B with ligand A. These asymmetric mutants are favored because they avoid producing ligands that clog or activate the receptors on their own surface. As a result, different types of cells are better at recognizing each other and mating more quickly.

When cells sexually reproduce they exchange genetic material with each other to produce offspring with a combination of genes that differ to their own. However, if the genes coding for ligand and receptor pairs were constantly being 'swapped', this could lead to new combinations, and a loss of asymmetric signaling. Hadjivasiliou and Pomiankowski showed that for asymmetric signaling to evolve, natural selection favors the genes encoding these non-compatible ligand and receptor pairs to be closely linked within the genome. This ensures that the mis-matching ligand and receptor are inherited together, preventing cells from producing pairs which can bind to themselves.

This study provides an original way to address an evolutionary question which has long puzzled biologists. These findings raise further questions about how gametes evolved to become the sperm and egg, and how factors such as signaling between cells can determine the sex of more complex organisms, such as ourselves.

DOI: https://doi.org/10.7554/eLife.48239.002

hypotheses have been studied extensively and offer compelling arguments. Nevertheless, the existence of several species where inbreeding (*Billiard et al., 2011*; *Perrin, 2012*) or biparental inheritance of mitochondria (*Billiard et al., 2011*; *Wilson and Xu, 2012*) are the rule but nonetheless maintain mating types, indicates that these ideas may not alone explain the evolution of mating types.

An alternative hypothesis is that mating types are determined by the molecular system regulating gamete interactions (*Hoekstra, 1987*; *Hadjivasiliou et al., 2015*; *Hadjivasiliou and Pomiankowski, 2016*). Such interactions dictate the success of mating by guiding partner attraction and recognition and the process of cell fusion, and have been shown to be more efficient when operating in an asymmetric manner (*Hadjivasiliou et al., 2015*). For example, diffusible molecules are often employed as signals that guide synchronous entry to gametogenesis or as chemoattractants (*Tsubo, 1961*; *Maier, 1993*; *Kuhlmann et al., 1997*; *Merlini et al., 2013*). Secreting and sensing the same diffusible molecule impedes the ability of cells to accurately detect external signals and makes partner finding many-fold slower (*Hadjivasiliou et al., 2015*). In addition, secreting and detecting the same molecule in cell colonies can prevent individuals responding to signals from others (*Youk and Lim, 2014*). Our previous review revealed that sexual signaling and communication in isogamous species are universally asymmetric (*Hadjivasiliou and Pomiankowski, 2016*). This applies throughout the sexual process from signals that lead to gametic differentiation, to attraction via diffusible pheromones and interactions via surface bound molecules during cell fusion (*Hadjivasiliou and Pomiankowski, 2016*).

In this work we take this analysis further by explicitly considering ligand-receptor interactions between and within cells. We directly follow the dynamics of ligand and receptor molecules that are

surface bound and determine the conditions under which the formation of within cell ligand-receptor pairs impedes between cell communication. We use this framework to explore the evolution of gametic interactions and show that asymmetric signaling roles and tight linkage between receptor and ligand loci both evolve due to selection for intercellular communication and quick mating. Our findings demonstrate that the evolution of mating type loci with suppressed recombination can be traced back to the fundamental selection for asymmetric signaling during sex.

## Theoretical set-up

Consider a population where cells encounter one another at random and can mate when in physical contact. Interactions between cells leading to successful mating are dictated by a ligand-receptor pair. Population wide effects may emerge if the ligand is highly diffusible (*Youk and Lim, 2014*; *Hadjivasiliou et al., 2015*). The employment of membrane bound ligands during sexual signaling is universal, whereas diffusible signals are not (*Hadjivasiliou and Pomiankowski, 2016*). In this work we therefore assume that the ligand-receptor interactions only operate locally. Receptors remain bound to the cell surface and ligands only undergo localized diffusion (*Figure 1*) as is the case in several yeast and other unicellular eukaryotes (*Cappellaro et al., 1991*; *Wilson et al., 1999*; *Phadke and Zufall, 2009*; *Merlini et al., 2013*). The following equations describe the concentration of free ligand $L$, free receptor $R$ and bound ligand $LR$ within a single cell,

$$\frac{d[L]}{dt} = \nu_L - k^+[R][L] + k^-[LR] - \gamma_L[L], \tag{1}$$

$$\frac{d[R]}{dt} = \nu_R - k^+[R][L] + k^-[LR] - \gamma_R[R], \tag{2}$$

$$\frac{d[LR]}{dt} = k^+[R][L] - k^-[LR] - \gamma_{LR}[LR]. \tag{3}$$

$\nu_L$ and $\nu_R$ describe the rate of production of the ligand and receptor respectively. $\gamma_L$, $\gamma_R$, and $\gamma_{LR}$, are the degradation rate of the ligand, receptor and bound complex respectively. The terms $k^+$ and $k^-$ are the binding and unbinding rates that determine the affinity of the ligand to its receptor within a single cell. We can solve *Equations (1-3)* by setting the dynamics to zero to obtain the amount of free ligand, free receptor ($[L]^*$, $[R]^*$) and bound complex at steady state ($[LR]^*$),

$$[L]^* = \frac{k^+\gamma_{LR}(\nu_L - \nu_R) - k^-\gamma_L\gamma_R - \gamma_L\gamma_R\gamma_{LR} + \Delta}{2k^+\gamma_L\gamma_{LR}} \tag{4}$$

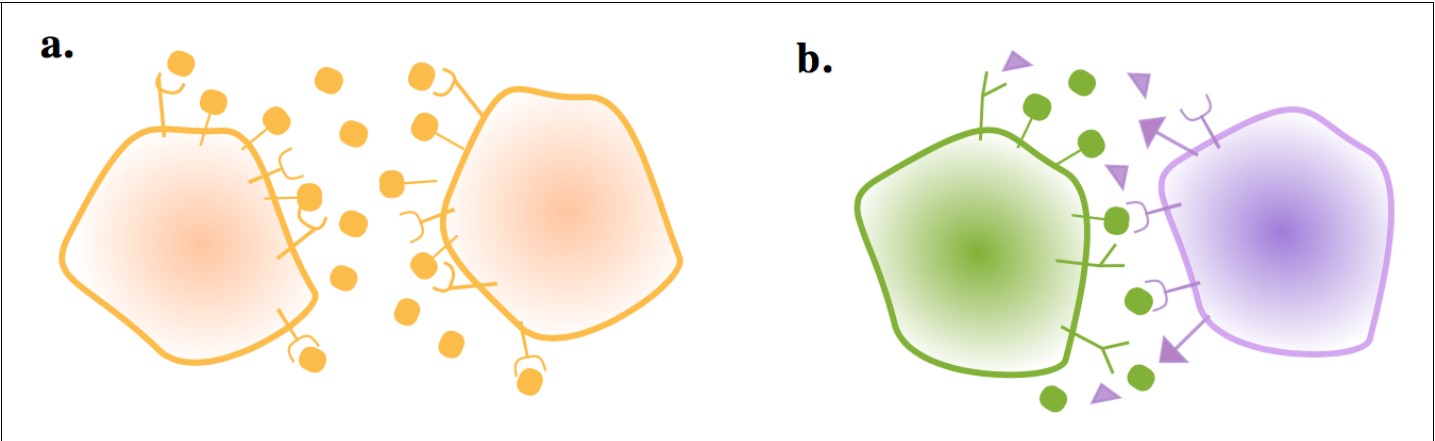

**Figure 1.** Gametes communicate through ligand and receptor molecules. The ligand can be either membrane bound or released in the local environment. (a) When the interacting cells produce ligand and receptor symmetrically, the ligand will bind to receptors on its own membrane as well as those on the other cell. This may impair intercellular signaling. (b) Producing the ligand and receptor in an asymmetric manner resolves this issue.
DOI: https://doi.org/10.7554/eLife.48239.003

$$[R]^* = \frac{k^+ \gamma_{LR}(\nu_R - \nu_L) - k^- \gamma_L \gamma_R - \gamma_L \gamma_R \gamma_{LR} + \Delta}{2k^+ \gamma_R \gamma_{LR}}, \tag{5}$$

$$[LR]^* = \frac{k^+ \gamma_{LR}(\nu_R + \nu_L) + k^- \gamma_L \gamma_R + \gamma_L \gamma_R \gamma_{LR} - \Delta}{2k^+ \gamma_{LR}^2}, \tag{6}$$

Where $\Delta$ is given by,

$$\Delta = \sqrt{(k^- \gamma_L \gamma_R + \gamma_{LR}(\gamma_L \gamma_R + k^+ \gamma_{LR}(\nu_R + \nu_L)))^2 + 4k^+ \gamma_L \gamma_R \gamma_{LR}(k^- + \gamma_{LR})\nu_R}. \tag{7}$$

We assume that the rates of ligand and receptor production and degradation are associated to timescales that are much shorter than the timescale of interactions between cells. Hence the concentrations of $[L]$, $[R]$ and $[LR]$ in individual cells will be at steady state when two cells meet. The likelihood of a successful mating between two cells depends not just on partner signaling levels but also on how accurately the cells can compute the signal produced by their partner. Binding of ligand and receptor originating from the same cell can obstruct this interaction. To capture this, we define the strength of the incoming signal for cell$_1$ when it interacts with cell$_2$ as,

$$W_{12} = k_b [L_2]^* [R_1]^* \left( 1 - \frac{[LR_1]^*}{[LR_1]^* + k_b [L_2]^* [R_1]^*} \right)^n, \tag{8}$$

where subscripts denote concentrations in cell$_1$ and cell$_2$, and the parameter $k_b$ determines the affinity of the ligand and receptor between cells. If $k_b$ is the same as the affinity of receptor and ligand within cells, then $k_b = \frac{k^+}{k^-}$. We also consider cases where $k_b \neq \frac{k^+}{k^-}$, for example, when ligand interacts differently with receptors on the same as opposed to a different cell (*LeBon et al., 2014*; *Hadjivasiliou et al., 2016a*).

The cost of self-signaling is determined by $n$. When $n = 0$, $W_{12}$ reduces to $k_b [R_1]^* [L_2]^*$ with the incoming signal dependent on the concentration of ligand produced by cell$_2$ and receptor produced by cell$_1$. This corresponds to a case where self-binding does not lead to activation but only causes an indirect cost through the depletion of available ligand and receptor molecules. When $n \geq 1$, binding within a cell leads to some form of activation that interferes with between cell signaling, imposing a cost in evaluating the incoming signal. Higher values for $n$ correspond to more severe costs due to self-binding.

The likelihood that two cells successfully mate ($P$) depends on the quality of their interaction given by,

$$P = \frac{W_{12} W_{21}}{K + W_{12} W_{21}}. \tag{9}$$

*Equation (9)* transforms the signaling interaction into a mating probability. For the analysis that follows, we choose large values of $K$ so that $P$ is far from saturation and depends almost linearly on the product $W_{12} W_{21}$. In summary, the probability that two cells mate is defined by the production and degradation rates of the ligand and receptor molecules, and the binding affinities between and within cells.

## Evolutionary model

To explore the evolution of signaling roles, we simplify the model by assuming that the degradation rates $\gamma_L, \gamma_R, \gamma_{LR}$ are constant and equal to $\gamma$, and investigate mutations that quantitatively modify the ligand and receptor production rates. We consider a finite population of $N$ haploid cells and set $N = 1000$ throughout the analysis unless otherwise stated. Ligand and receptor production are controlled by independent loci with infinite alleles (*Tajima, 1996*). The ligand and receptor production rates of cell$_i$ is denoted by $(\nu_{L_i}, \nu_{R_i})$. We also consider different versions of the ligand and its receptor. Cells have two ligand-receptor pairs, $(L, R)$ and $(l, r)$ which are mutually incompatible, so the binding affinity is zero between $l$ and $R$, and between $L$ and $r$. Each cell has a $(L, R)$ and $(l, r)$ state, which are subject to mutational and evolutionary pressure as described below. $W_{12}$ is re-defined as the summation of the interactions of these two ligand-receptor pairs,

$$W_{12} = k_b [L_2]^* [R_1]^* \left( 1 - \frac{[LR_1]^*}{[LR_1]^* + k_b [L_2]^* [R_1]^*} \right)^n + k_b [l_2]^* [r_1]^* \left( 1 - \frac{[lr_1]^*}{[lr_1]^* + k_b [l_2]^* [r_1]^*} \right)^n . \tag{10}$$

Again for the sake of simplicity, the ligand-receptor affinities are set to be the same between and within cells for each ligand-receptor pair (i.e. $k^+$, $k^-$ and $k_b$ are the same for $L - R$ and $l - r$ interactions). A cell undergoes recurrent mutation that changes the production rate for the ligand $L$ so that $\nu'_{L_i} = \nu_{L_i} + \epsilon$ with $\epsilon \sim N(0, \sigma)$ with probability $\mu$. The same mutational process occurs for all ligand and receptor production rates. We assume that mutation occurs independently at different loci and that there is a maximum capacity for ligand and receptor production, so that $\nu_L + \nu_l < 1$ and $\nu_R + \nu_r < 1$. It follows that the production rates of the two ligand genes are not independent of one another and similarly for the two receptor genes.

We also consider cases where $\nu_L + \nu_l < \alpha$ and $\nu_R + \nu_r < \alpha$ for $\alpha \neq 1$ to reflect the relative synergy ($\alpha > 1$) or relative competition ($\alpha < 1$) between the production of the two ligands (or receptors). For example, synergy between two ligands (or receptors) could reflect reduced energy expenditure for the cell if the same machinery is used to produce the two molecules. Competition on the other hand could reflect additional costs due to the production of two different ligands (or receptors).

Selection on ligand-receptor production rates is governed by the likelihood that cells pair and produce offspring. We assume that cells enter the sexual phase of their life cycle in synchrony, as is the case in the majority of unicellular eukaryotes (*Hadjivasiliou and Pomiankowski, 2016*). Pairs of cells are randomly sampled (to reflect random encounters) and mate with probability $P$ defined in *Equation (9)*. Cells failing to mate are returned to the pool of unmated individuals. The process is repeated until $M$ cells have mated, giving rise to $M/2$ mated pairs (we set $M < N$, so only some cells mate). Each mated pair produces 2 haploid offspring so the population size shrinks from $N$ to $M$. The population size is restored back to $N$ by sampling with replacement. It follows that *Equations (9) and (10)* together provide a proxy for fitness according to the ligand and receptor production rates of individual cells. Initially, recombination is not allowed between the genes controlling ligand and receptor production but then is considered in a later section.

## Results

### Dependence of gamete interactions on physical parameters

The strength of an incoming signal $W_{12}$ depends on the concentration of free receptor in cell$_1$ and free ligand in cell$_2$, and the cost of self-binding ($n$) (*Equation (10)*). The steady state concentration of $[L]$, $[R]$ and $[LR]$ are governed by different production rates (*Figure 2—figure supplement 1*; details of the derivation can be found in the Materials and methods section). For low degradation rates ($\gamma$ small), the removal of available molecules is dominated by self-binding ($k^+$) (*Equations (1) and (2)* and *Figure 2a,b*). At the same time, a lower degradation rate leads to higher levels of ligand and receptor (*Figure 2a*) even if the relative drop of free ligand and receptor is steeper as $k^+$ increases (*Figure 2b*). As a consequence, the ability of a cell to generate a strong signal and read incoming signals can change drastically when the pair of interacting cells produce the ligand and receptor in a symmetric manner (e.g. $(\nu_L, \nu_R, \nu_l, \nu_r) = (1, 1, 0, 0)$ for both cells) rather than in an asymmetric manner (e.g. $(\nu_{L_1}, \nu_{R_1}, \nu_{l_1}, \nu_{r_1}) = (1, 0, 0, 1)$ and $(\nu_{L_2}, \nu_{R_2}, \nu_{l_2}, \nu_{r_2}) = (0, 1, 1, 0)$). The fold-increase in $W_{12}$ is large even when self-binding confers no cost ($n = 0$), while larger values for $n$ ramp up the costs (*Figure 2c*). If cells produce the ligand and receptor asymmetrically, self-binding ceases to be a problem in receiving incoming signals.

Although the strength of the signaling interaction between two cells ($W_{12} W_{21}$) may improve when the interacting cells produce the ligand and receptor asymmetrically, this need not be the case. Consider the interaction of a resident cell with production rates $(\nu_L, \nu_R, \nu_l, \nu_r)_{res} = (1, 1, 0, 0)$ with itself and a mutant cell with production rates given by $(\nu_L, \nu_R, \nu_l, \nu_r)_{mut} = (1 - dx, 1 - dy, dx, dy)$. For all values of $dx$ and $dy$, $[W_{12} W_{21}]_{res+mut} - [W_{12} W_{21}]_{res+res} < 0$ (*Figure 3a*). It follows that $(\nu_L, \nu_R, \nu_l, \nu_r) = (1, 1, 0, 0)$ cannot be invaded by any single mutant.

However, if the resident is already slightly asymmetric, for example $(\nu_L, \nu_R, \nu_l, \nu_r)_{res} = (1, 0.9, 0, 0.1)$, then a mutant conferring an asymmetry in the opposite direction can be better at interacting with the resident (*Figure 3b*). When the resident produces both ligand and

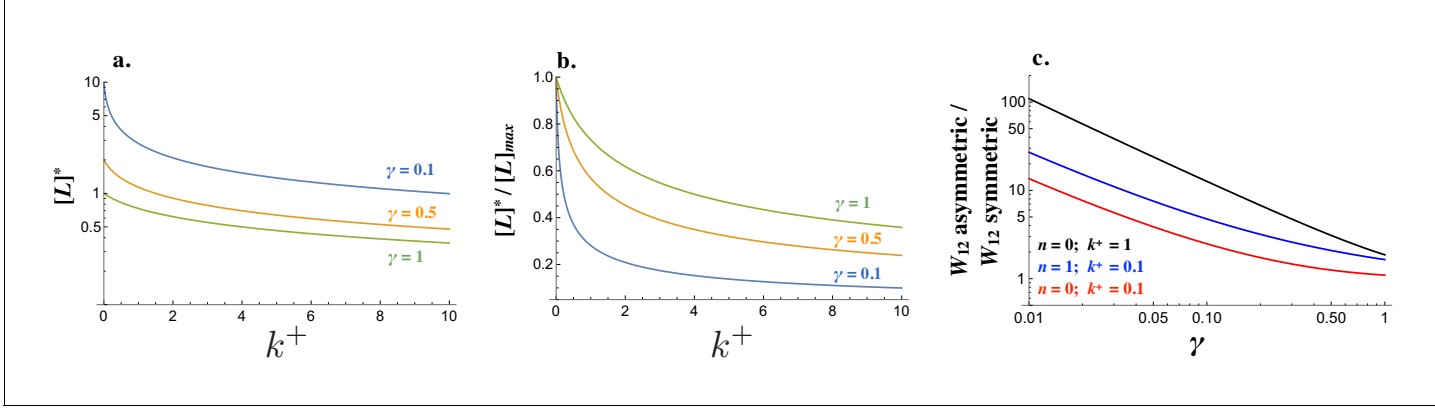

**Figure 2.** Signaling interactions between mating cells can be severely impaired due to ligand-receptor interactions in the same cell. (a) The amount of free ligand in individual cells at steady state $[L]^*$ and (b) normalized amount of free ligand at steady state $[L]^*/[L]_{max}$ varies with the intracellular binding rate $k^+$ and degradation rate $\gamma$. (c) The relative amount of incoming signal $W_{12}$ for a cell that produces ligand and receptor asymmetrically versus symmetrically decreases with the degradation rate $\gamma$ and values of intracellular binding $k^+$. Other parameters used: $n = 1, k^- = 1, k_b = 1$.

DOI: https://doi.org/10.7554/eLife.48239.004

The following figure supplement is available for figure 2:

**Figure supplement 1.** Steady state concentrations in individual cells.

DOI: https://doi.org/10.7554/eLife.48239.005

receptor equally (e.g. $(\nu_L, \nu_R, \nu_l, \nu_r)_{res} = (0.5, 0.5, 0.5, 0.5)$; *Figure 3c*), then most mutants conferring an asymmetry in either ligand or receptor production are favored. The strongest interaction occurs with mutants that produce the ligand or receptor fully asymmetrically (i.e. $(\nu_L, \nu_R, \nu_l, \nu_r)_{mut} = (1, 0, 0, 1)$ or $(0, 1, 1, 0)$; (*Figure 3c*)). Finally, when the resident production rates are already strongly asymmetric given by $(\nu_L, \nu_R, \nu_l, \nu_r)_{res} = (1, 0, 0, 1)$, a mutant with an asymmetry in the opposite direction is most strongly favored (*Figure 3d*). Note that a population composed only of cells with production rates at $(\nu_L, \nu_R, \nu_l, \nu_r)_{res} = (1, 0, 0, 1)$ is not viable since the probability that two such cells mate is zero. However, this analysis provides insight about how asymmetry in signaling evolves.

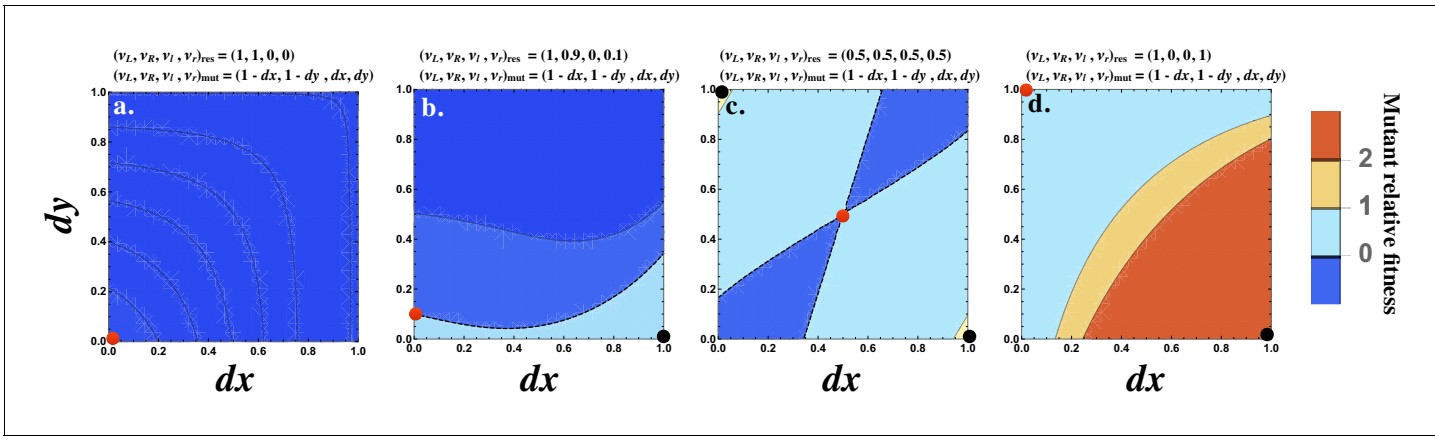

**Figure 3.** Fitness advantage of rare mutations conferring signaling asymmetry. The fitness of a rare mutant is plotted relative to the resident $[W_{12}W_{21}]_{res+mut} - [W_{12}W_{21}]_{res+res}$. The production rate of the mutant cell is $(\nu_L, \nu_R, \nu_l, \nu_r)_{mut} = (1 - dx, 1 - dy, dx, dy)$, where $dx$ and $dy$ are plotted on the $x$ and $y$ axes respectively. The resident production rate $(\nu_L, \nu_R, \nu_l, \nu_r)_{res}$ is shown as a red dot and varies (a) $(1, 1, 0, 0)_{res}$, (b) $(1, 0.9, 0, 0.1)_{res}$, (c) $(0.5, 0.5, 0.5, 0.5)_{res}$ and (d) $(1, 0, 0, 1)_{res}$. The mutant $(dx, dy)$ with maximum fitness is shown as a black dot. The contour where $[W_{12}W_{21}]_{res+mut} = [W_{12}W_{21}]_{res+res}$ is marked by a black dashed line (b and c). The fitness difference is always negative in (a) and always positive in (d). Other parameters used: $n = 1, \gamma = 0.5, k^+ = 1, k^- = 1, k_b = 1$.

DOI: https://doi.org/10.7554/eLife.48239.006

## Evolution of mating types with asymmetric signaling roles

To explore the evolution of signaling asymmetry, we follow mutations that alter the relative production of two mutually incompatible types of ligand and receptor $(L, R)$ and $(l, r)$. To ease understanding, the population symmetry $s$ in the production of ligand and receptor is measured,

$$s = 1 - \frac{1}{2N}\sum_{i=1}^{N}(|\nu_{L_i} - \nu_{R_i}| + |\nu_{l_i} - \nu_{r_i}|). \tag{11}$$

The population is symmetric ($s = 1$) if cells produce ligand and receptor equally, for both types (i.e. $(\nu_R, \nu_{L_i}$, for constant $a$), and fully asymmetric ($s = 0$) when cells adopt polarized roles (i.e. $(\nu_L, \nu_R, \nu_l, \nu_r) = (1, 0, 0, 1)$ and $(0, 1, 1, 0)$).

Starting from a population where all cells are symmetric producers of only one ligand and receptor, $(\nu_L, \nu_R, \nu_l, \nu_r) = (1, 1, 0, 0)$, the population evolves to one of two equilibria (**Figure 4a**). $E_1$ where $s^* \approx 1$ and all cells produce the ligand and receptor symmetrically $(\nu_L, \nu_R, \nu_l, \nu_r) \approx (1, 1, 0, 0)$ or $E_2$ where $s^* \approx 0$ and the population is divided into ligand and receptor producing cells, with equal frequencies of $(\nu_L, \nu_R, \nu_l, \nu_r) \approx (1, 0, 0, 1)$ and $(\nu_L, \nu_R, \nu_l, \nu_r) \approx (0, 1, 1, 0)$ (**Figure 4b,c**). Equilibria with intermediate values of $s^*$ are not found. The exact production rates at $E_1$ and $E_2$ exhibit some degree of noise due to mutation and finite population size (**Figure 4b,c**). At $E_2$, individual cells with high $\nu_R$ (and low $\nu_r$) have low $\nu_L$ (and high $\nu_l$), confirming that $s^* \approx 0$ captures a fully asymmetric steady state (**Figure 4b,c**).

Whether $E_2$ is reached from $E_1$ depends on key parameters that determine the strength of self-binding and signaling interactions between cells. $E_1$ persists and no asymmetry evolves when $k^+$ (the intracellular ligand-receptor binding coefficient) is small (**Figure 4d**). In this case, the concentration of self-bound ligand-receptor complex is small (**Equation (6)**) and there is little cost of self-signaling (**Equation (8)**), so there is weak selection in favor of asymmetry. When the population is at $E_1$, asymmetric mutants are slightly deleterious on their own (**Figure 3a**). They are therefore more likely to be lost when $k^+$ is small and selection for asymmetric signaling is weak (**Figure 4d**). The opposite is true for larger values of $k^+$, as self-binding now dominates and restricts between cell signaling, promoting the evolution of asymmetry (**Figure 4d**). The transition from $E_1$ to $E_2$ occurs at a smaller value of $k^+$ when the degradation rate ($\gamma$) is decreased (**Figure 4d**), as the effective removal of free ligand and receptor depends more strongly on intercellular binding (**Figure 2a,b**). Furthermore, the mutation rate affects the value of $k^+$ at which the transition from $E_1$ to $E_2$ occurs. The transition from $E_1$ to $E_2$ when mutation rates are smaller occurs at larger $k^+$ (**Figure 4—figure supplement 1**). We further explore the role of the mutational process below.

Another important consideration is the relative strength of signaling within and between cells, given by $k^+/k^-$ and $k_b$ respectively. For example, the threshold value of the within cell binding rate beyond which symmetric signaling ($E_1$) evolves to asymmetric signaling ($E_2$, **Figure 4a**) increases when $k_b$ becomes much larger than $k^+/k^-$ (**Figure 4e**). Furthermore, this threshold value is smaller for larger values of $n$ indicating that asymmetric signaling is more likely to evolve when the cost for self-signaling is higher (larger $n$, **Figure 4e**). However, asymmetric signaling can evolve even when self-binding carries no cost ($n = 0$) as high rates of self-binding can restrict the number of ligand and receptor molecules free for between cell interactions (**Figure 4e**).

We also wondered how the relative synergy or competition between the two ligands (or receptors) could affect our results. When the two ligands (or receptors) exhibit synergy so that $\nu_L + \nu_l < \alpha$ and $\nu_R + \nu_r < \alpha$ for $\alpha > 1$, a signaling asymmetry evolves more easily (for smaller values of $k^+$, **Figure 4—figure supplement 2**). Now the second ligand (or receptor) begins to evolve without imposing a cost on the preexisting ligand (or receptor) and can therefore remain present in the population longer until an asymmetry in the opposite direction evolves in other cells. The reverse dynamics are observed when the two ligands (or receptors) compete with one another ($\nu_L + \nu_l < \alpha$ and $\nu_R + \nu_r < \alpha$ for $\alpha < 1$ ) (**Figure 4—figure supplement 2**).

The observations above suggest that both $E_1$ and $E_2$ are evolutionary stable states and the transition from $E_1$ to $E_2$ depends on the mutational process, drift and the parameters that determine signaling interactions. To explore this we investigated the stability of $E_1$ in response to rare mutations in the receptor and ligand production rates. We assume the population is initially at $E_1$ (i.e. $(\nu_L, \nu_R, \nu_l, \nu_r) = (1, 1, 0, 0)$), introduce mutations in the receptor and ligand loci $(\nu_L, \nu_R, \nu_l, \nu_r) = (1 - dx, 1, dx, 0)$ and $(\nu_L, \nu_R, \nu_l, \nu_r) = (1, 1 - dy, 0, dy)$ at frequency $p$, and calculate the

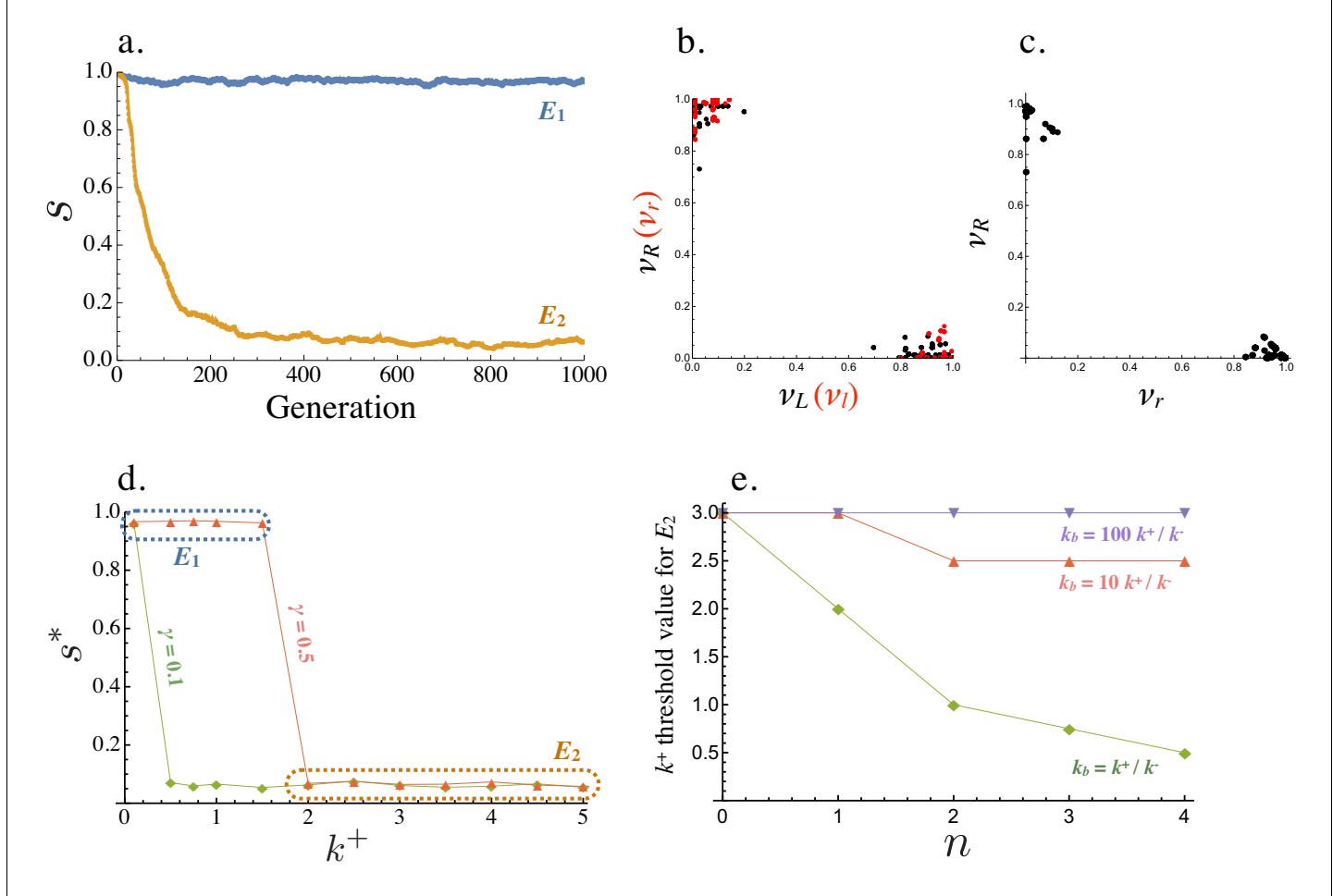

**Figure 4.** Evolution of asymmetric signaling. (a) An example of evolution to the two signaling equilibria, $E_1$ ($s = 1$ full symmetry when $k^+ = 1$) and $E_2$ full asymmetry when $k^+ = 5$). (b) Production rates of individual cells in the population for the receptor-ligand pairs $L - R$ (black) and $l - r$ (red) at $E_2$. (c) Production rates of individual cells for the two receptor types $R$ and $r$ at $E_2$. (d) Steady state signaling symmetry $s^*$ against the intracellular binding rate ($k^+$) for different degradation rates ($\gamma$). (e) Threshold value of $k^+$, beyond which $E_2$ evolves from $E_1$, plotted versus the cost of self-binding ($n$). The relationship is shown for different values of strength of between cell signaling ($k_b$) relative to strength of within cell signaling ($k^+/k^-$). Other parameters used in numerical simulations are given in the Supplemental Material.

DOI: https://doi.org/10.7554/eLife.48239.007

The following figure supplements are available for figure 4:

**Figure supplement 1.** The role of mutation rates.
DOI: https://doi.org/10.7554/eLife.48239.008
**Figure supplement 2.** Synergy and competition between the production rates of the two ligands (and receptors).
DOI: https://doi.org/10.7554/eLife.48239.009

population symmetry at steady state for different values of $dx$ and $dy$ (**Figure 5**). Single mutations never spread (i.e. if $dx = 0$ no value of $dy$ allows mutants to spread and vice versa). This is in agreement with the analytical predictions presented in the previous section (**Figure 3a**). When both $dx$ and $dy$ are nonzero the population may evolve to $E_2$, where the two mutants reach equal frequencies at ~0.5 and replace the resident. The basin of attraction for $E_2$ (and so asymmetric signaling roles) is larger when $k^+$ and $p$ are high and $\gamma$ is small (**Figure 5a–d**), as predicted analytically (**Figures 2** and **3**) and in accordance with our findings when mutations were continuous (**Figure 4**).

Note that the initial mutation frequency ($p$) matters in our system. Single mutations are slightly deleterious on their own as predicted analytically (**Figure 3a**) and seen here when $dx = 0$ or $dy = 0$ (**Figure 5**). The two mutants, however, can be favored when they are asymmetric in opposite directions (i.e. $dx > 0$ and $dy > 0$; **Figure 5**). When mutants are introduced at a lower frequency (compare

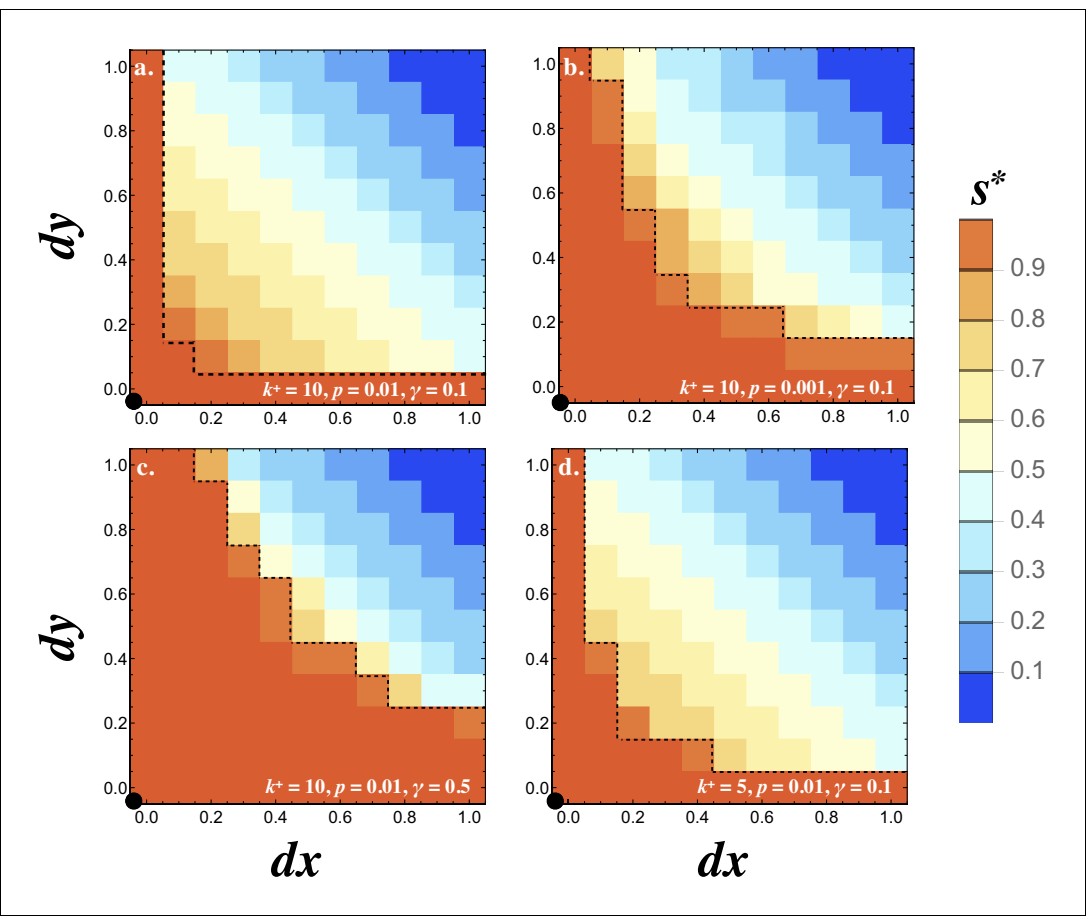

**Figure 5.** Invasion of $E_1$. Contour plots showing the steady state degree of symmetry ($s^*$) in a population with
resident $(\nu_R, \nu_L, \nu_r, \nu_l) = (1, 1, 0, 0)$. Two mutations are introduced $(1 - dx, 1, dx, 0)$ and $(1, 1 - dy, 0, dy)$ at frequency
$p$ and their fate is followed until they reach a stable frequency. No recurrent mutation is assumed. Orange
contours inside the dotted line show the region where both mutants are eliminated and the resident persists
($s^* = 1$). All other colors indicate that the two mutants spread to equal frequency 0.5 displacing the resident
($s^*{<}1$).
The degree of signaling symmetry at equilibrium is dictated by the magnitude of the mutations given by $dx$ and
$dy$. The different panels show (a) between cell signaling $k^+ = 10$, mutation frequency $p = 0.01$ and degradation rate
$\gamma = 0.1$, (b) lower mutation frequency $p = 0.001$, (c) high degradation rate $\gamma = 0.5$ and (d) weaker between cell
signaling $k^+ = 5$. The resident type is marked by a black dot at the origin. The frequency of the resident and two
mutants at steady state was recorded and the heat maps show the average steady state value of $s^*$ for 20
independent repeats and the population size $N = 10000$. Other parameters used and simulation details are given
in the Supplementary Material.
DOI: https://doi.org/10.7554/eLife.48239.010

*Figure 5a–b*), the probability that they meet one another before they are lost by drift increases. This
explains why smaller values of $p$ result in narrower basins of attraction for $E_2$ (*Figure 5a–b*).

We next investigated how mutations invade when the resident already signals asymmetrically (i.e.
produces both ligands). The resident was set to $(\nu_L, \nu_R, \nu_l, \nu_r)_{res} = (1 - dx, 1, dx, 0)$ and a mutant able
to produce both receptors $(\nu_L, \nu_R, \nu_l, \nu_r)_{mut} = (1, 1 - dy, 0, dy)$ was introduced. If $dx{>}0$, a mutant con-
veying a small asymmetry in receptor production (i.e. $dy{>}0$) increases in frequency until the popula-
tion reaches a polymorphic state with the resident and mutant at 50% (*Figure 6a*). If $dx{>}0$ but the
mutant only produces one receptor (i.e. $dy = 0$), the mutant invades, reaching a low frequency when
$dx$ is small and replaces the resident when $dx$ is large. It follows that an asymmetry in both ligand
and receptor production is necessary for the evolution of a signaling asymmetry as predicted analyti-
cally (*Figure 3a*).

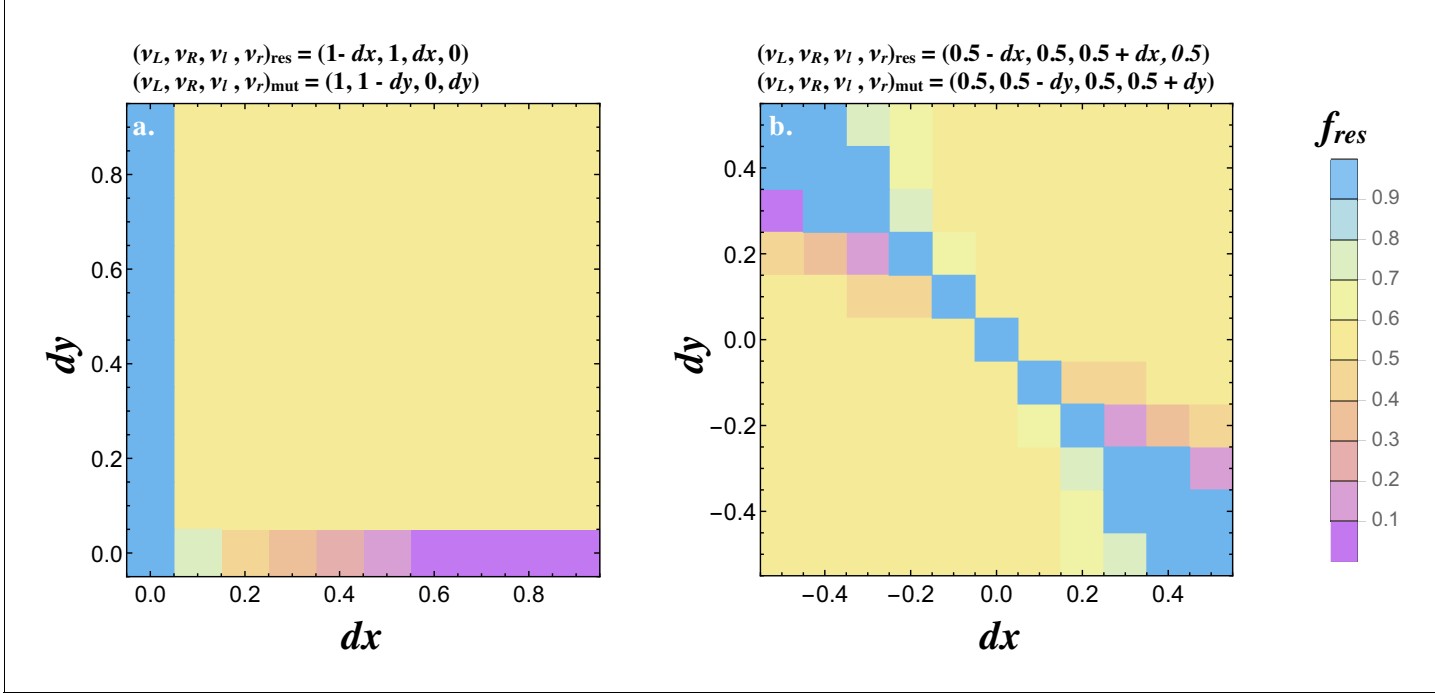

**Figure 6.** Joint evolution of receptor and ligand asymmetry. Contour plots show the equilibrium frequency of the resident ($f_{res}$) following the introduction of a mutant when (a) $(\nu_L, \nu_R, \nu_l, \nu_r)_{res} = (1 - dx, 1, dx, 0)$ and, $(\nu_L, \nu_R, \nu_l, \nu_r)_{mut} = (1, 1 - dy, 0, dy)$, and (b) $(\nu_L, \nu_R, \nu_l, \nu_r)_{res} = (0.5 - dx, 0.5, 0.5 + dx, 0.5)$ and $(\nu_L, \nu_R, \nu_l, \nu r)_{mut} = (0.5, 0.5 - dy, 0.5, 0.5 + dy)$. The mutant is introduced at a frequency $p = 0.01$ and no recurrent mutation is assumed. Other parameters used and simulation details are given in the Supplemental Material.
DOI: https://doi.org/10.7554/eLife.48239.011

We also consider a resident type that produces both ligands and both receptors with some degree of asymmetry in ligand production (i.e. $(\nu_L, \nu_R, \nu_l, \nu_r)_{res} = (0.5 - dx, 0.5, 0.5 + dx, 0.5)$) and map the spread of a mutant with asymmetry is receptor production $(\nu_L, \nu_R, \nu_l, \nu_r)_{mut} = (0.5, 0.5 - dy, 0.5, 0.5 + dy)$. The pairwise invasibility plots for values of $dx$ and $dy$ show that signaling asymmetries in opposite directions are favored. These evolve to a polymorphic state with equal frequencies of cells at $dx = dy = -0.5$ and $dx = dy = 0.5$ (**Figure 6b**). These findings together illustrate how the asymmetric state $E_2$ evolves from the symmetric state $E_1$.

## Effects of recombination

The results above assume that the loci controlling ligand and receptor production are tightly linked which prevents the production of deleterious combinations following meiosis. Recombination is a minor problem at the $E_1$ equilibrium which is monomorphic (except for mutational variation). But it is likely to be a problem at the polymorphic $E_2$ equilibrium. For example, at $E_2$ mating between $(\nu_L, \nu_R, \nu_l, \nu_r) = (1, 0, 0, 1)$ and $(0, 1, 1, 0)$ cells generates non-asymmetric recombinant ligand-receptor combinations, either $(1, 1, 0, 0)$ or $(0, 0, 1, 1)$. To implement recombination we assume that the two ligands are tightly linked in a single locus and are inherited as a pair (likewise the two receptors), and investigate the effects of recombination between the ligand locus and the receptor locus. Note that if we allow recombination between ligands (or receptors), this would be expected to generate combinations with a similar deleterious impact.

Consider the effect of recombination on a population at $E_1$. As before, the population either stays at $E_1$ or evolves to $E_2$ dependent on parameter values (**Figure 7a**). When the population evolves to $E_2$, $s^*$ becomes larger as the recombination rate, $(\rho)$, increases (**Figure 7b**). For low recombination rates ($\rho \leq 0.1$), the population largely consists of equal frequencies of $(1, 0, 0, 1)$ and $(0, 1, 1, 0)$ cells, producing the ligand and receptor asymmetrically. A small percentage of recombinant cells produce conspecific pairs of ligand and receptor $(\nu_L, \nu_R, \nu_l, \nu_r) = (1, 1, 0, 0)$ and $(0, 0, 1, 1)$ (**Figure 7b,c**). Recombination in this case creates 'macromutations' where production rates that were 0 become 1

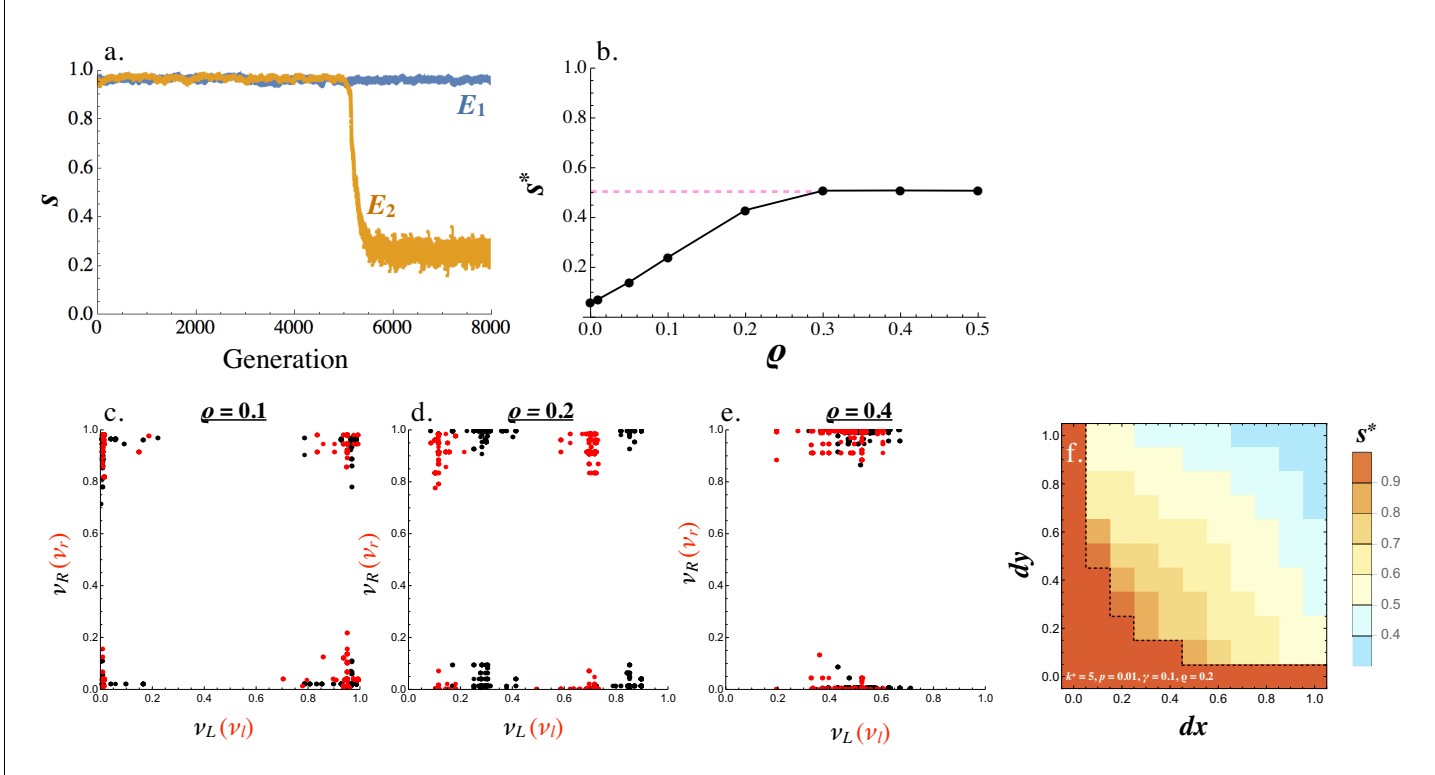

**Figure 7.** The effect of recombination on $E_2$. (a) An example of evolution of the two signaling equilibria, $E_1$ (for $k^+ = 1$) and $E_2$ (for $k^+ = 5$) given a fixed recombination rate $\rho = 0.1$. (b) Steady state $s^*$ varies with the recombination rate. (c–d) Production rates of individual cells in the population for receptor-ligand pairs $L - R$ (black) and $l - r$ (red) for recombination rates (c) $\rho = 0.1$, (d)$\rho = 0.2$ and (e) $\rho = 0.4$. (f) Contour plot showing the steady state degree of symmetry ($s^*$) in a population with resident $(\nu_R, \nu_L, \nu_r, \nu_l) = (1, 1, 0, 0)$, given a recombination rate $\rho = 0.2$. Two mutations are introduced $(1 - dx, 1, dx, 0)$ and $(1, 1 - dy, 0, dy)$ at rate $p$ and their fate is followed until they reach a stable frequency. The population size $N = 1000$ for panels (a) - (e) and $N = 10000$ for panel (f). Other parameters used and simulation details are given in the Supplemental Material.
DOI: https://doi.org/10.7554/eLife.48239.012

and vice versa. As the recombination rate rises ($\rho \geq 0.2$), the two leading cell types diverge from $(\nu_L, \nu_R, \nu_l, \nu_r) = (1, 0, 0, 1)$ and $(0, 1, 1, 0)$ towards $(1 - \epsilon_1, \epsilon_2, \epsilon_3, 1 - \epsilon_4)$ and $(\epsilon_5, 1 - \epsilon_6, 1 - \epsilon_7, \epsilon_8)$ where the $\epsilon_i$ are below 0.5 but greater than zero *Figure 7d*). Higher recombination rates ($\rho \geq 0.3$) push $s^* = 0.5$ at $E_2$ (*Figure 7b*). Here, there is a predominance of $(\nu_L, \nu_R, \nu_l, \nu_r) = (1, 0.5, 0, 0.5)$ and $(0, 0.5, 1, 0.5)$ cells at equal frequencies (or $(0.5, 1, 0.5, 0)$ and $(0.5, 0, 0.5, 1)$ by symmetry). This arrangement is robust to recombination since the receptor locus is fixed at $(\nu_R, \nu_r) = (0.5, 0.5)$ and the ligand locus is either at $(\nu_L, \nu_l) = (1, 0)$ or $(0, 1)$ (or $(\nu_L, \nu_l) = (0.5, 0.5)$) and the receptor is either at $(\nu_R, \nu_r) = (1, 0)$ or $(0, 1)$). So pairing between these two cell types results in $(1, 0.5, 0, 0.5)$ and $(0, 0.5, 1, 0.5)$ offspring, whether recombination occurs or not. Note that this arrangement maintains some degree of asymmetry even with free recombination ($\rho = 0.5$). Even though both cell types produce both receptors, they produce the ligand asymmetrically (or vice versa). Cells on average are more likely to mate successfully between rather than within the two types of cells.

Similar to the case of no recombination, the invasion of $E_1$ by $E_2$ depends on the mutational process and parameter values. *Figure 7f* shows the steady state symmetry measure in a population initially at $(\nu_L, \nu_R, \nu_l, \nu_r) = (1, 1, 0, 0)$ when two mutations $(1 - dx, 1, dx, 0)$ and $(1, 1 - dy, 0, dy)$ are introduced at low frequencies. Whether or not the mutants invade depends on the magnitude of the mutation in a similar way as in the case of no recombination (*Figure 5d* versus *Figure 7f*). However, the value of $s^*$ now diverges from zero reflecting the nonzero rate of recombination.

## Evolution of linkage

In the analysis above, recombination between the ligand and receptor loci is fixed. However, the recombination rate itself can evolve. To investigate this, we let the recombination rate $\rho$ undergo

recurrent mutation with probability $\mu_\rho$ so that the mutant recombination rate becomes $\rho' = \rho + \varepsilon_\rho$ with $\varepsilon_\rho \sim N(0, \sigma_\rho)$. In a diploid zygote, the rate of recombination is given by the average of the two recombination alleles, $\rho_1$ and $\rho_2$, carried by the mating cells. In this way, the recombination rate evolves together with the ligand and receptor production rates. We start with maximal recombination rate $\rho = 0.5$ and $(v_L, v_R, v_l, v_r) = (1, 1, 0, 0)$ for all cells and allow the recombination rate to evolve by drift for 1000 generation before we introduce mutation in the ligand and receptor loci.

The recombination rate evolves to $\rho^* = 0$ whenever $E_2$ was reached from $E_1$ in the no recombination analysis. Under these conditions, tight linkage between receptor and ligand genes is favored (**Figure 8a**). Furthermore, asymmetric signaling roles coevolve together with the recombination rate. The evolved trajectories of $s$ and $\rho$ depend on the strength of selection for asymmetric signaling. For example, when $k^+$ is large ($k^+ = 10$), signal asymmetry rapidly evolves; $s$ moves away from one and this is followed by a sharp drop in the recombination rate (**Figure 8b**). Eventually the population evolves asymmetric signaling roles ($s$ in orange, **Figure 8b**) and tight linkage ($\rho$ in blue, **Figure 8b**). These dynamics are similar when $k^+$ is smaller ($k^+ = 3$, **Figure 8c**) and selection for

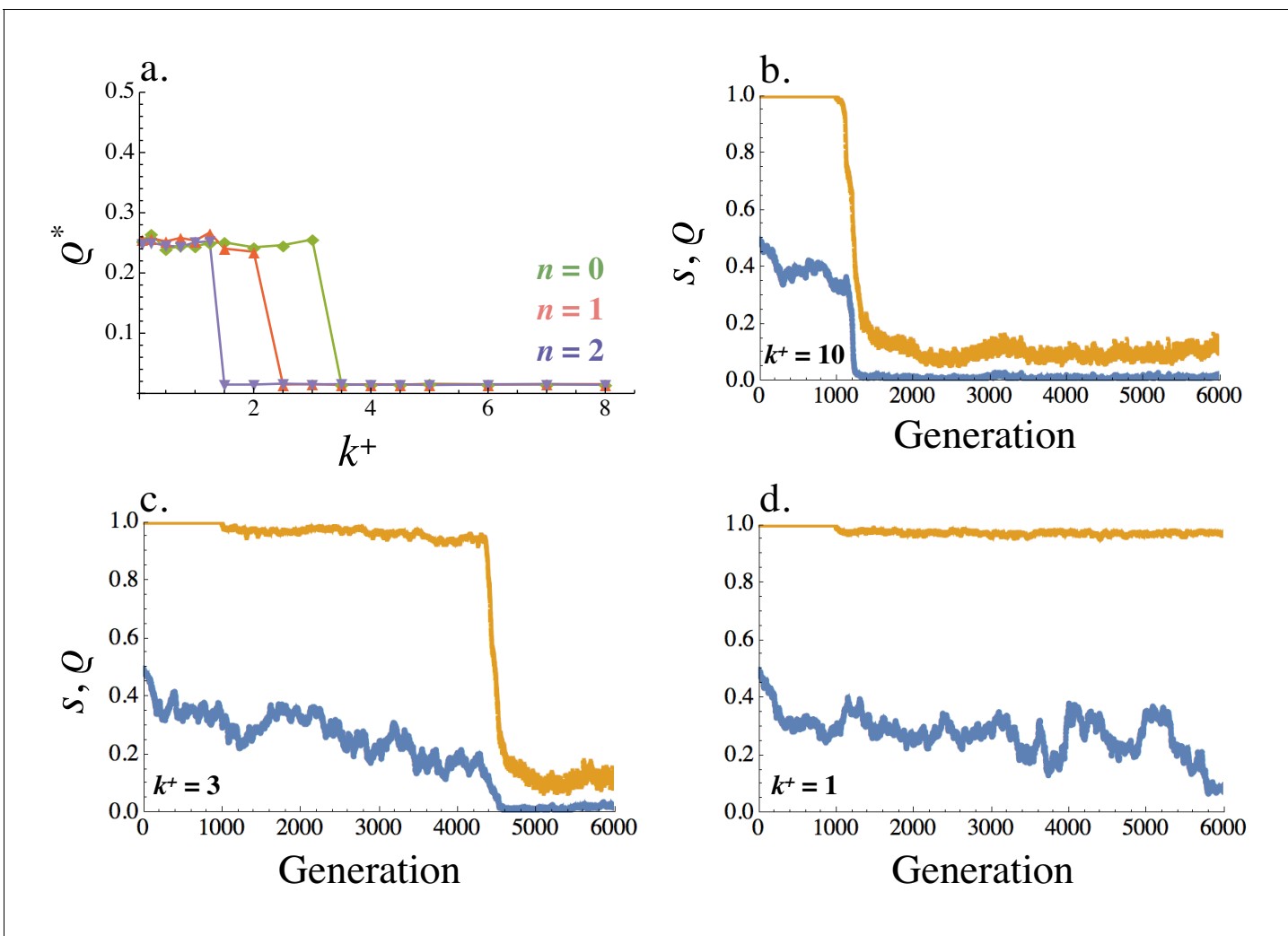

**Figure 8.** Equilibrium recombination rate $\rho^*$. (a) Averaged across the population, $\rho^*$ varies with $k^+$ (within cell binding rate) and $n = 0, 1, 2$ (cost of self-binding). (b–d) Evolution of the recombination rate $\rho$ (blue) and signaling symmetry levels $s$ (orange) for different within cell binding rates: (b) $k^+ = 10$, (c) $k^+ = 3$ and (d) $k^+ = 1$. The recombination rate evolves under drift for the first 1000 generations, following which mutation at the ligand and receptor loci were introduced. When no asymmetry evolves the recombination rate fluctuates randomly between 0 and 0.5 (i.e. between its minimum and maximum value like a neutral allele). Other parameters used in simulations are given in the Supplemental Material.
DOI: https://doi.org/10.7554/eLife.48239.013

asymmetry is weaker. However, it now takes longer for the asymmetric types to co-evolve (*Figure 8c*). When selection for asymmetric signaling is even weaker ($k^+ = 1$, *Figure 8d*), no asymmetry evolves ($s$ remains at 1) and the recombination rate fluctuates randomly between its minimum and maximum value as one would expect in the case of a neutral allele.

## Discussion

Explaining the evolution of mating types in isogamous organisms constitutes a major milestone in understanding the evolution of anisogamy and sexes (*Randerson and Hurst, 2001*; *Lehtonen et al., 2016*). Mating type identity is determined by a number of genes that reside in regions of suppressed recombination and code for ligands and receptors that guide partner attraction and recognition, as well as genes that orchestrate cell fusion and postzygotic events (*Billiard et al., 2011*; *Perrin, 2012*; *Hadjivasiliou and Pomiankowski, 2016*; *Branco et al., 2018*). In this work we show that an asymmetry in ligand and receptor production evolves as a response to selection for swift gamete communication and mating. Furthermore, the same conditions favoring asymmetric signaling select for tight linkage between the receptor and ligand genes. Our findings indicate that selection for asymmetric signaling roles could have played an important role in the early evolution of gamete differentiation and identity.

We investigated the evolution of mating type roles by considering two types of ligand and receptor in individual cells. Gene duplication followed by mutation is a well established route to novelty evolution (*Susumu, 1970*; *Zhang, 2003*; *Magadum et al., 2013*) and could explain the co-existence of two pairs of ligand and receptor in our system. Alternatively, individual cells could produce multiple ligands and receptors which evolve independently, as is the case in some basidiomycete fungi (*Fowler and Vaillancourt, 2007*). The production rate of the two types of ligand (and receptor) in our system is subject to mutation using an assumption of infinite alleles (*Tajima, 1996*), so that the amount of expressed ligand (and receptor) of each kind is modulated quantitatively. In this way we were able to explicitly express the likelihood of mating as a function of the amount of free and bound molecules on the cell membrane and the ability of cells to accurately read their partner's signal. This framework allowed us to follow the evolution of the quantitative production of ligand and receptor in mating cells for the first time.

We found that the ligand-receptor binding rate within a cell ($k^+$) is key in the evolution of asymmetric signaling roles (*Figures 3* and *4*). $k^+$ holds an important role because it dictates the rate at which free ligand and receptor molecules are removed from the cell surface. In addition, $k^+$ determines the amount of intracellular signal that interferes with the ability of cells to interpret incoming signal. Although in theory cells could avoid self-binding (by reducing $k^+$ to zero), there is likely to be a strong association of the within-cell and between-cell binding affinities. So reductions in $k^+$ are likely to have knock-on costs in reducing $k_b$ as well. An extreme example is the case of locally diffusible signals (*Figure 1*), such as those used by ciliates and yeasts to stimulate and coordinate fusion (*Sugiura et al., 2010*; *Merlini et al., 2013*). Here binding affinities between and within cells are inevitably identical (since the ligand is not membrane bound). Work in yeast cells has shown that secreted ligands utilized for intercellular signaling during sex are poorly read by cells that both send and receive the same ligand (*Youk and Lim, 2014*). In the case of strictly membrane bound molecules avoiding self-binding could also be an issue as it requires a ligand and receptor pair that bind poorly within a cell without compromising intercellular binding. For example, choosy budding yeast gametes (which are better at discriminating between species) take longer to mate (*Rogers et al., 2015*). It would be interesting to further study these trade-offs experimentally.

We never observed the co-existence of a symmetric 'pansexual' type with asymmetric self-incompatible types. The two steady states consist of either a pansexual type alone or two mating types with asymmetric signaling roles. This could explain why the co-existence of mating types with pansexuals is rare in natural populations (*Billiard et al., 2011*; *Billiard et al., 2012*). This is in contrast to previous models where pansexual types were very hard to eliminate due to negative frequency dependent selection (*Hoekstra, 1982*; *Czárán and Hoekstra, 2004*; *Hadjivasiliou et al., 2013*). For example, in the case of the mitochondrial inheritance model, uniparental inheritance raises fitness not only in individuals that carry genes for uniparental inheritance but also for pansexual individuals (benefits 'leak' to biparental individuals) (*Hadjivasiliou et al., 2013*; *Christie and Beekman, 2017b*).

A similar pattern is seen with inbreeding avoidance because the spread of self-incompatibility reduces the population mutation load, and so reduces the need for inbreeding avoidance (*Czárán and Hoekstra, 2004*). These dynamics are reversed in the present model where there is positive frequency dependent selection. The spread of asymmetric signalers generates stronger selection for further asymmetry (*Figures 3* and *4*). This also occurs when there is recombination (*Figures 7* and *8*). Even though recombination between the two asymmetric types generates symmetric recombinant offspring, these are disfavored and eliminated by selection. These observations suggest that the mitochondrial inheritance and inbreeding avoidance models are unlikely to generate strong selection for suppressed recombination which is the hallmark of mating types. Finally, it would be interesting to explore how the reinstatement of recombination could be a route back to homothallism which is a state derived from species with mating types (*Billiard et al., 2011*).

Mating type identity in unicellular eukaryotes is determined by mating type loci that typically carry a number of genes (*Billiard et al., 2012*; *Hadjivasiliou and Pomiankowski, 2016*). Suppressed recombination at the mating type locus is a common feature across the evolutionary tree (*Branco et al., 2018*). Our work predicts the co-evolution of mating type specific signaling roles and suppressed recombination with selection favoring linkage between loci responsible for signaling and an asymmetry in signaling roles. This finding suggests that selection for asymmetric signaling could be the very first step in the evolution of tight linkage between genes that control mating type identity. In yeasts, the only genes in the mating type locus code for the production of ligand and receptor molecules (*Merlini et al., 2013*). These then trigger a cascade of other signals downstream that also operate asymmetrically. Evidence across species suggests that mating type loci with suppressed recombination are precursors to sex chromosomes (*Menkis et al., 2008*; *Geng et al., 2014*). In this way our work provides crucial insights about the origin of sex chromosomes.

The framework developed here could be used together with recent efforts to understand numerous features of mating type evolution. For example, opposite mating type gametes often utilize diffusible signals to attract partners (*Luporini et al., 1995*; *Tsuchikane et al., 2005*). The inclusion of long range signals such as those used in sexual chemotaxis will provide further benefits for asymmetric signaling roles and mating types (*Hadjivasiliou et al., 2015*). Furthermore the number of mating types varies greatly across species and is likely to depend on the frequency of sexual reproduction and mutation rates (*Constable and Kokko, 2018*). Signaling interactions between gametes could also play a role in determining the number of mating types and reducing their number to only two in many species (*Hadjivasiliou et al., 2016b*). It would be interesting to use the framework developed here to study the evolution of additional ligands and receptors and their role in reaching an optimal number of mating types. Other important features such as the mechanism of mating type determination (*Billiard et al., 2011*; *Vuilleumier et al., 2013*) and stochasticity in mating type identity (*Hadjivasiliou et al., 2016b*; *Nieuwenhuis and Immler, 2016*; *Nieuwenhuis et al., 2018*) could also be understood in light of this work.

Our analysis revealed that the evolution of asymmetric gamete signaling and mating types is contingent upon the mutation rate. Single mutants that exhibit an asymmetry are initially slightly disadvantageous. When further mutations emerge that are asymmetric in the opposite direction, a positive interaction between these mutants occurs that can lead to the evolution of distinct mating types. When the population size is small and mutation rates are low, there is a low probability that individuals carrying asymmetric mutations in opposite directions are segregating at the same time. Increasing the population size or the mutation rate would enhance the probability of co-segregation, making the evolution of asymmetric signaling more likely. In an infinite population the evolution of signaling asymmetry should be independent of the mutation rate. Finally, it is worth noting that unicellular eukaryotes undergo several rounds of asexual growth (tens to thousands) between each sexual reproduction (*Hadjivasiliou et al., 2016b*; *Constable and Kokko, 2018*). It follows that the effective mutation rate between sexual rounds will end up being orders of magnitude higher than the mutation rates at each vegetative step.

Taken together our findings suggest that selection for swift and robust signaling interactions between mating cells can lead to the evolution of self-incompatible mating types determined at non-recombinant mating type loci. We conclude that the fundamental selection for asymmetric signaling between mating cells could be the very first step in the evolution of sexual asymmetry, paving the way for the evolution of anisogamy, sex chromosomes and sexes.

## Materials and methods

### General model

We model $N$ cells so that each cell is individually characterized by a ligand locus $\mathcal{L}$ and a receptor locus $\mathcal{R}$. Two ligand genes at the locus $\mathcal{L}$ determine the production rates for two ligand types $l$ and $L$ given by $\nu_l$ and $\nu_L$. Similarly, two receptor genes at the locus $\mathcal{R}$ determine the production rates for the two receptor types $r$ and $R$ given by $\nu_r$ and $\nu_L$. The two ligand and receptor genes in our model could could arise from duplication followed by mutation that leaves two closely linked genes that code for different molecules. In our computational set-up each cell is associated with production rates $\nu_l$, $\nu_L$, $\nu_r$ and $\nu_R$ where we assume a normalized upper bound so that $\nu_l + \nu_L < 1$ and $\nu_r + \nu_R < 1$.

The steady state concentrations for $L, R,$ and $LR$ are computed by setting $\frac{d[L]}{dt} = \frac{d[R]}{dt} = \frac{d[LR]}{dt} = 0$ in *Equations (1-3)* and solving the resulting quadratic equations. This leads two solutions only one of which gives positive concentrations. It follows that there is a unique physical solution to our system, which is what we use to define the probability of mating in our numerical simulations.

The program is initiated with $\nu_L = \nu_R = 1$ and $\nu_l = \nu_r = 0$ for all cells (unless otherwise stated, see Section 5.4). We introduce mutation so that the ligand and receptor production rates of individual cells mutate independently with probability $\mu$. A mutation event at a production gene changes the production rate by an increment $\epsilon$ where $\epsilon \sim N(0, \sigma)$. Mutation events at the different genes $l, L, r$ and $R$ are independent of one another. If $\nu_l + \nu_L > 1$ or $\nu_l + \nu_L > 1$ the production rates are renormalized so their sum is capped at 1. If a mutation leads to a production rate below 0 or above one it is ignored and the production rate does not change.

We implement mating by randomly sampling individual cells. The probability that two cells mate is determined by their ligand and receptor production rates as defined in *Equation (9)* in the main text. We assume that $K$ takes a large value relative to $W_{12}W_{21}$ so that $P$ is linear in $W_{12}W_{21}$. Because the absolute value for $W_{12}W_{21}$ varies greatly between parameter sets, and what we are interested in is the relative change in $W_{12}W_{21}$ when signaling levels change, we chose $K$ to be equal to the maximum value $W_{12}W_{21}$ can take for a given choice of $\gamma$, $k^+$, $k^-$ and $k_b$. Sampled cells that do not mate are returned to the pool of unmated cells. This process is repeated until $M = N/2$ cells have successfully mated. This produces $N/4$ pairs of cells each of which gives rise to two offspring. These are sampled with replacement until the population returns to size $N$. We assume that a mutation-selection balance has been reached when the absolute change in $s$, defined in *Equation (10)* in the main text, between time steps $t_1$ and $t_2$ is below $\epsilon = 10^{-5}$ across $t_2 - t_1 = 100$. Certain parameter sets resulted in noisy steady states and were terminated following $10^5$ generations. The numerical code keeps track of all production rates for individual cells over time.

### Adaptive dynamics

We model adaptive dynamics by initiating the entire population at state $(\nu_L, \nu_R, \nu_l, \nu_r)_{res}$ and introducing a mutant $(\nu_L, \nu_R, \nu_l, \nu_r)_{mut}$ at low frequency $p$. We allow the population to evolve according to the life cycle introduced in the main text and record the frequency of the resident and mutant type when a steady state is reached. For the purposes of *Figure 5*, the resident type is set to $(\nu_L, \nu_R, \nu_l, \nu_r)_{res}$ and two mutants $(\nu_L, \nu_R, \nu_l, \nu_r)_{mut_1}$ and $(\nu_L, \nu_R, \nu_l, \nu_r)_{mut_2}$ are introduced both at frequency $p$. In this case we track the frequencies of the resident and both mutants until steady state is reached. We define steady state as the point where the average value of $s$ in the population between time steps $t_1$ and $t_2$ is below $\epsilon = 10^{-7}$ across $t_2 - t_1 = 100$. The population always reached steady state.

### Recombination

We implement recombination by considering a modifier $\mathcal{M}$ that lies between the ligand and receptor loci $\mathcal{L}$ and $\mathcal{R}$. That is, we assume that the two ligand genes and two receptor genes are tightly linked on the ligand and receptor locus $\mathcal{L}$ and $\mathcal{R}$ respectively, and only model recombination between the two loci. For simplicity, we assume that $\mathcal{M}$ determines the physical distance between $\mathcal{L}$ and $\mathcal{R}$ so that the distances $\mathcal{L} - \mathcal{M}$ and $\mathcal{R} - \mathcal{M}$ are the same. The modifier $\mathcal{M}$ determines the rate of recombination between the ligand and receptor loci quantitatively by determining $\rho_M$, the probability of a single recombination event following mating. Consider for example two individuals whose

ligand and receptor production rates and recombination rates are determined by the triplets $R_1 - M_1 - L_1$ and $R_2 - M_2 - L_2$, the possible offspring resulting from such a mating are given by,

1. $R_1 - M_1 - L_1$ and $R_2 - M_2 - L_2$ with probability $(1 - \rho_{M_{1,2}})^2$ – equivalent to no recombination events
2. $R_1 - M_2 - L_1$ and $R_2 - M_1 - L_2$ with probability $\rho_{M_{1,2}}^2$ – equivalent to two recombination events
3. $R_1 - M_2 - L_2$ and $R_2 - M_1 - L_1$ with probability $\rho_{M_{1,2}}(1 - \rho_{M_{1,2}})$ – equivalent to one recombination event
4. $R_1 - M_1 - L_2$ and $R_2 - M_2 - L_1$ with probability $\rho_{M_{1,2}}(1 - \rho_{M_{1,2}})$ – equivalent to one recombination event

where $\rho_{M_{1,2}} = \frac{1}{2}(\rho_{M_1} + \rho_{M_2})$ is the joint recombination rate when cell$_1$ and cell$_2$ with recombination rates $\rho_{M_1}$ and $\rho_{M_2}$ respectively mate.

We allow mutation at the recombination locus at rate $\mu_\rho$ independently of the ligand and receptor loci. A mutation event leads to a new recombination rate so that $\rho'_M = \rho_M - \epsilon$ for $\epsilon \sim N(0, \sigma_\rho)$. We assume that the mutation-selection balance has been reached when the absolute change in $s$, defined in *Equation (10)* in the main text, and the change in the average recombination rate between time steps $t_1$ and $t_2$ is below $\epsilon = 10^{-5}$ across $t_2 - t_1 = 100$.

## Methods and parameters used for simulated figures

### Figure 4
(a): Individual simulations following the trajectory of $s$ over time. Population is initiated at $(\nu_L, \nu_R, \nu_l, \nu_r) = (1, 1, 0, 0)$ and $\rho = 0$ for all cells at time 0. $\mu = 0.01$ for all ligand and receptor genes and $\mu_r = 0$, $\gamma = 0.1$, $k^- = 1$ , $n = 1$, $k_b = k^+/k^-$ and $k^+ = 1$ for $E_1$ trajectory and 5.0 for $E_2$ trajectory. Population size $N = 1000$ and number of cells allowed to mate $M = N/2$.

(b-c): Parameters as for (a) with $k^+ = 5.0$. Each dot is represents an individual cell in the simulation.

(d): Parameters used as for (a) with varying $k^+$ and $\gamma$ as indicated in the Figure. Simulation was run until a steady state was reached and the value of $s^*$ was averaged over the last 1000 time steps to account for noise.

(e): Parameters used as for (a), varying $k_b$ and $n$ as indicated in the Figure. $k^+$ was also varied here and the value of $k^+$ beyond which $E_2$ evolved at the expense of $E_1$ was noted (the y-axis value).

### Figure 5
Adaptive dynamics simulations following the frequency of two mutants $(\nu_L, \nu_R, \nu_l, \nu_r) = (1 - dx, 1, dx, 0)$ and $(\nu_L, \nu_R, \nu_l, \nu_r) = (1, 1 - dy, 0, dy)$ introduced at frequency $p$ (indicated on Figure ) in a resident population with $(\nu_L, \nu_R, \nu_l, \nu_r) = (1, 1, 0, 0)$. The frequency of the resident and two mutants at steady state was recorded and the heat maps show the average steady state value of $s^*$ for 20 independent repeats. Parameters used: $\gamma = 0.5$, $k^- = 1$, $n = 1$, $k_b = k^+/k^-$, $N = 10000$, $M = N/2$.

### Figure 6
Joint evolution of receptor and ligand asymmetry. Contour plots show the equilibrium frequency of the resident ($f_{res}$) with production rates $(\nu_L, \nu_R, \nu_l, \nu_r)_{res} = (1 - dx, 1, dx, 0)$ (a) $(\nu_L, \nu_R, \nu_l, \nu_r)_{res} = (0.5 - dx, 0.5, 0.5 + dx, 0.5)$ (b), following a mutation $(\nu_L, \nu_R, \nu_l, \nu_r)_{mut} = (1, 1 - dy, 0, dy)$ (a) and $(\nu_L, \nu_R, \nu_l, \nu_r)_{mut} = (0.5, 0.5 - dy, 0.5, 0.5 + dy)$ (b). The mutant is introduced at a frequency $p = 0.001$. Other parameters used and simulations details are given in the Supplemental Material.

### Figure 7
(a): Individual simulations following the trajectory of $s$ over time. Population is initiated at $(\nu_L, \nu_R, \nu_l, \nu_r) = (1, 1, 0, 0)$ and $\rho = 0.1$ for all cells at time 0. $\mu = 0.01$ for all ligand and receptor genes and $\mu_r = 0$. $\sigma = 0.1$, $\gamma = 0.5$, $k^- = 1$, $n = 1$, $k_b = k^+/k^-$ and $k^+ = 1$ for $E_1$ trajectory and 5.0 for $E_2$ trajectory. Population size $N = 1000$ and number of cells allowed to mate $M = N/2$.

(b): Parameters as in (a) but varying $\rho$ as indicated in the Figure and using $k^+ = 3.0$. The $y$ axis shows the steady state value of $s$ averaged over 1000 steps after steady state has been reached.

(c-e): Parameters as for (a) with $k^+ = 5.0$ and recombination rate $\rho$ as shown in each Figure . Each dot is represents an individual cell in the simulation.

(f): Parameters as for (a) with $k^+ = 5$, $\mu_b = 0.01$, $\rho = 0.2$ and $N = 10000$. The heat maps show the value of $s^*$ at steady state averaged over 20 repeats. Heat map was obtained in the same way as *Figure 5*.

## Figure 8

(a): Population is initiated at $(\nu_L, \nu_R, \nu_l, \nu_r) = (1, 1, 0, 0)$ and $\rho = 0.5$ for all cells at time 0. $\mu = 0.01$ for all ligand and receptor genes and $\mu_\rho = 0.01$, $\gamma = 0.5$, $k^- = 1$, $k_b = k^+/k^-$ and $n$ vary as shown in the plot. The y axis shows the steady state value of $\rho$ averaged over 1000 steps after steady state has been reached. Population size $N = 1000$ and number of cells allowed to mate $M = N/2$.

(b-d): Parameters as in (a) with $k^+$ varied as shown in the individual plots.

## Acknowledgements

This research was funded by an Engineering and Physical Sciences Research Council Fellowship (EP/L50488/) and HFSP Long Term Fellowship to ZH, and by grants from the Engineering and Physical Sciences Research Council (EP/F500351/1, EP/I017909/1, EP/K038656/1) and the Natural Environment Research Council (NE/R010579/1) to AP. The authors thank Michael Doebeli and three anonymous reviewers for comments that have helped improve this manuscript. ZH would like to thank Natalia Tokarova for her work on an earlier version of this project as a summer student at UCL, and Marcos Gonzalez-Gaitan for his support.

## Additional information

### Funding

| Funder | Grant reference number | Author |
| --- | --- | --- |
| Human Frontier Science Program | Long Term Fellowship | Zena Hadjivasiliou |
| Engineering and Physical Sciences Research Council | EP/L50488/ | Zena Hadjivasiliou |
| Engineering and Physical Sciences Research Council | EP/F500351/1 | Andrew Pomiankowski |
| Natural Environment Research Council | NE/R010579/1 | Andrew Pomiankowski |
| Engineering and Physical Sciences Research Council | EP/I017909/1 | Andrew Pomiankowski |
| Engineering and Physical Sciences Research Council | EP/K038656/1 | Andrew Pomiankowski |

The funders had no role in study design, data collection and interpretation, or the decision to submit the work for publication.

### Author contributions

Zena Hadjivasiliou, Conceptualization, Data curation, Formal analysis, Funding acquisition, Validation, Investigation, Visualization, Methodology, Writing—original draft, Writing—review and editing; Andrew Pomiankowski, Data curation, Formal analysis, Funding acquisition, Investigation, Writing—review and editing

### Author ORCIDs

Zena Hadjivasiliou https://orcid.org/0000-0003-1174-1421
Andrew Pomiankowski https://orcid.org/0000-0002-5171-8755

Decision letter and Author response
Decision letter https://doi.org/10.7554/eLife.48239.016
Author response https://doi.org/10.7554/eLife.48239.017

## Additional files

### Supplementary files

• Transparent reporting form
DOI: https://doi.org/10.7554/eLife.48239.014

### Data availability

Code for simulations was written in C++ and the simulated data was analysed using Mathematica. C++ code and Mathematica notebooks are available on GitHub (https://github.com/zenah12/SignalingMatingTypes; copy archived at https://github.com/elifesciences-publications/SignalingMatingTypes).

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
