## [Decision Letter]

[Editors’ note: a previous version of this study was rejected after peer review, but the authors submitted for reconsideration. The first decision letter after peer review is shown below.]

Thank you for submitting your work entitled "Evolution of the mating type locus with suppressed recombination" for consideration by *eLife*. Your article has been reviewed by a Senior Editor, a Reviewing Editor, and three reviewers. The reviewers have opted to remain anonymous.

Our decision has been reached after consultation between the reviewers. Based on these discussions and the individual reviews below, we regret to inform you that we have decided to reject your paper in its present form. However, we would be willing to consider a revised version of your manuscript that takes into account the extensive reviewer comments.

The most important point in such a revision would be to address the criticism that it is unclear in the present version of the manuscript what exactly leads to the evolution of the mating type equilibrium E2. In particular, it is unclear whether the analytical and the simulation results presented in the paper lead to the same conclusions, and how the evolution of the asymmetric state depends on the mutational process. The relationship between the analytical and the simulation results needs to be made much clearer, and both reviewers 1 and 3 suggest doing an invasion analysis to clarify this point, e.g. using adaptive dynamics and pairwise in viability plots.

Other points you may wish to consider are the evolution of more than two mating types, as well as the possible effects of population-wide ligand and receptor production.

*Reviewer #1:*

Here the authors ask if self-incompatible mating types evolve from a self-compatible ancestor lacking mating types as a result of selection to improve cross-cell signalling. The intuition is clear; by producing a signal different from that being received from potential partners there is less self-interference, and this selects for two different mating types (one sending A and receiving B, one sending B and receiving A). This intuition is demonstrated with a ligand-receptor (signal-receiver) model, in which an individual can produce two different pairs of ligands and receptors. These molecules interact to form bound complexes, these complexes break down, and all molecules and complexes decay (equations 1-3). From this the equilibrium concentrations of receptors, ligands, and complexes are calculated (equations 4-7) and used to determine the probability of mating given meeting (equation 9) as a function of the signal strengths between the two cells (equation 10). The population is updated, from one generation to the next, by randomly sampling pairs of cells, until M pairs have mated. The offspring of these pairs are sampled with replacement until a population size of N is recovered. Genetic variation arises by mutation, inducing a small random normal deviation, at each of the 4 genes controlling the production rates of the 2 ligands and 2 receptors, independently (with some renormalization to keep the total ligand and receptor production below a maximum). The authors first show how the signal strength (and as a result, fitness) depends on the trait value (ligand-receptor production rates) of both the focal individual and the potential mating partner (Figure 2). In particular, populations producing only one pair of ligand-receptor molecules are evolutionary stable, but as more individuals produce the other ligand or receptor (e.g., by drift) there can be selection for individuals producing the ligand or receptor that matches this, suggesting the evolution of mating types might be an alternative stable evolutionary state. The authors then perform simulations to investigate the characteristics of this mating type equilibrium and the factors allowing it (Figure 3), as well as how the rate of recombination between the ligand and receptor loci affect this equilibrium (Figure 4). Finally, the authors then allow the recombination rate itself to evolve, showing that there is strong selection for reduced recombination at the mating type equilibrium (Figure 5). Thus, the model describes a hypothesis for the evolution of mating types with non-recombining mating type loci from an ancestor without mating types.

I do not know the literature well enough to assess the novelty of this paper. I enjoyed reading it and thought it thought provoking. As far as I can tell the analysis is correct (I did not check the simulations).

Essential revisions:

I think a bit more could be done to demonstrate why the mating type equilibrium (E2) evolves from the monomorphic ancestor (E1), which I think is unclear in a few places (details below). One way to do this would be to take the approach of adaptive dynamics, and assume a monomorphic population (e.g., all individuals make only ligand 1 and receptor 1, *v* = (1,1,0,0)). You then calculate the fitness of a mutant with trait value *v*' = (*x,y*,1-*x*,1-*y*) in this population, take the derivatives with respect to *x* and *y*, and evaluate at the resident equilibrium (*v*' = *v*). A phase plane can then be shown, illustrating the direction of selection acting on small mutations around the resident. Immediately from this one sees that E1 is locally stable (as the current Figure 2D suggests), as are the other four corners of the phase space. Of particular interest are the basins of attraction for each equilibrium, the effect of parameter values on the size of these basins, and the effects of parameter values on the strength of selection keeping a population within each basin (the "depth" of the basin if you will). The authors have rightly stated that there are multiple basins of attraction (subsection “Evolution of mating types with asymmetric signaling roles”) but in other places claim that E2 evolves from E1 because of strong selection for asymmetry (subsection “Evolution of mating types with asymmetric signaling roles”), implying E1 is instead unstable (i.e., there is no basin of attraction for E1). This idea of instability is implicit in Figure 3D-E, where the idea is that there are threshold values of *k^+^*, γ, and n that make E1 unstable. I think that the phase plane demonstrates that E1 remains stable for *k^+^* finite and γ positive (I've only looked at n=0). Instead of changing the local stability of E1 these parameters change the size of the basin of attraction and its depth, therefore affecting the rate of drift out of E1 (so that "threshold" values of *k^+^*, γ, and n are just the values that allow this rate to be high enough to see E1 lost by the maximum number of generations simulated). In particular, increasing *k^+^* decreases E1's basin of attraction and also makes the basin shallower, greatly facilitating drift out. In contrast, decreasing γ shrinks E1's basin of attraction but makes it deeper, meaning the rate at which E1 transitions to E2 might be less sensitive to changes in γ. This selection surface approach brings a number of new insights (e.g., increasing the size of basin vs. increasing the depth of the basin, the possibility of calculating the amount of drift needed to leave E1), could be used to explain the quantitative patterns in Figure 3D-F (e.g., yellow in Figure 3F is the boundary of the basin of attraction of E1), and highlights that E1 is always a stable state (which is not always clear from the text; subsection “Evolution of mating types with asymmetric signaling roles”). Further investigations along these lines could help illustrate the location and properties of the branching point leading to E2 (e.g., is there are "garden of eden" branching point at *v* = (0.5, 0.5, 0.5, 0.5)?). Additionally, one might take a quantitative genetic approach and use, for example, diffusion equations to ask what the rate of drift out of E1 is (e.g., Barton and Rouhani, 1987).

*Reviewer #2:*

This paper investigates the early evolution of the architecture of mating type locus, where individuals are initially bearing two pairs of ligand-receptors genes. The authors show that asymmetry in ligand-receptors production rates and linkage between genes are expected under a wide variety of conditions. The fundamental underlying mechanisms are the propensity of individuals to successfully mate with other individuals, which depends on their propensity to bind others' ligands on their own receptors, i.e. properly receiving a good signal. This propensity is modeled following a system of equations derived from biochemical reaction systems.

The authors investigate in particular the evolution of a population where mutations affect production rates of receptors and ligands. The model is analysed with computer individual-based simulations in a finite population with a fixed size.

In my opinion, this paper is excellent, especially because the model presented is very simple, and yet, it explains a lot. This model explains (1) why individuals can have an advantage in providing asymmetrical receptor-ligand production, (2) under which biochemical conditions should asymmetry evolve, (3) to what extent linkage between genes can affect the evolution of asymmetry, and (4) that recombination between genes is expected to evolve to zero (tight linkage) as asymmetry evolves to maximum. Such a model can apply to many different taxa: ciliates, fungi, algae, where receptor-ligand asymmetry in a mating type locus without recombination is common. Because it is very simple, the model is certainly very general. These results provide a first principle for the evolution of asymmetry between two kinds of individuals in a single population, which could trigger further differentiation for instance anisogamy or sexual chromosomes.

My only criticism would be that there can be a contradiction between two hypotheses: diffusion of ligands is local, and the use of differential equations with products between molecular concentrations as the rate of contact. This precludes the possibility that ligands produced by the whole population might interact and disturb the signalling between cells. Considering interactions between only two close cells would need more precise modeling of spatialized and stochastic processes. I am not sure that it would change a lot of the results, but I can imagine that considering the whole population ligands production could affect a lot the evolutionary outcomes since it might affect a lot the importance of the self-binding parameter *k^+^*. It would have been interesting to compare whole population vs. two cells interactions.

*Reviewer #3:*

Summary:

The authors study the evolution of unicellular mating types, where mating types are defined in terms of their ligand-receptor signalling toolkit. Each cell can produce two different types of receptors and ligands. Cells reproduce faster more compatible cells, i.e. ligands and receptors, there are in the population. They find that when self-binding rate is not small, evolution favours "asymmetric signalling". That is, half of the cells in the population start producing only one receptor and ligand-type, which are of opposite type to avoid self-binding, while in the other half of the population it is the other way round. I find the paper very clearly written, the research interesting and worthy of publication. I do however have few concerns and comments that must be addressed before I can recommend this paper for publication.

Essential revisions:

1) How does the evolution towards asymmetry kick off?a) It seems that for either of the two initially non-utilized receptors "r" or ligands "l" to have a selective advantage, the other must already be present in the population. Is this true?

b) If true, is it that you can nevertheless get evolution towards asymmetry because the mutation rates are assumed high and so at the mutation-selection balance there are always some "*l*" and "*r*" in the population? Would you ever observe evolution towards E2 if mutation rates would be of several orders lower?

c) To gain more insight on the evolution towards asymmetry, can you perform some (analytical) invasion analysis e.g. check the stability of E1 and E2?

2) It could be helpful to include some simple phase-plane/stability analysis for the model (1-3), at least in the appendix. How do we know the concentration of *L* and *R* reach their steady state (4-6)? Also, is this the only steady state? Also, in (1-3) should it be *dt* instead of *d[R*]?

3) Subsection “Theoretical set-up: Should the timescale separation argument be the other way round? Because if cells encounter each other at high rate, they’re within cell concentration of *L* and *R* would not have enough time to reach their steady state. Why is this a reasonable assumption?

4) Is there a reason for not assuming trade-off in *v_L_* vs. *v_R_*? One could imagine that cell can't arbitrarily increase the production of both *L* and *R*. Also, wouldn't such a trade-off alleviate the conditions for the evolution of asymmetry?

5) If you would allow for three different receptors and ligands per cell, would three mating types evolve? Using the same logic, would you get an arbitrary number of mating types if the population would be sufficiently large and the mutation rates sufficiently high?

[Editors’ note: what now follows is the decision letter after the authors submitted for further consideration.]

Thank you for submitting your article "Evolution of asymmetric gamete signaling and suppressed recombination at the mating type locus" for consideration by *eLife*. Your article has been reviewed by Detlef Weigel as the Senior Editor, a Reviewing Editor, and two reviewers. The reviewers have opted to remain anonymous.

The reviewers have discussed the reviews with one another and the Reviewing Editor has drafted this decision to help you prepare a revised submission. The consensus was that this is essentially ready to be accepted for publication, provided that the few points are addressed. The revised submission will not need to be sent out for review again.

*Reviewer #1:*

Thanks to the authors for carefully addressing my comments. I'm glad to see some of my ideas implemented, which I think add to the intuition gained here. I think there is still a bit of care needed in describing the stability of E1 and E2 (both comments in subsection “Evolution of mating types with asymmetric signaling roles”and Discussion section). I also think some care is needed in using "epistasis" to describe frequency dependent selection (subsection “Evolution of mating types with asymmetric signaling roles”), and analogies to fitness valley crossing might be useful.

*Reviewer #3:*

In my opinion the authors adequately addressed all the points raised by the reviewers. I recommend this paper for publication, provided the two points below will be taken into account.

1) It would be good to mention in the "main" text the population size that was used in the simulations. For example, in Figure 5 where you discuss the different mutation rates, in my opinion the population size should also be present. This leads me to a more general comment that it should be made clearer that the mutation rates used throughout the manuscript are unusually high (or at least I failed to find any mention on this caveat.). My guess is that if they would be decreased below the rates used in the paper, say to 10^-4^ or 10^-6^, the evolution to asymmetry would become extremely difficult, hinting that perhaps something else is going on in the evolution of gamete signalling in addition to what is already discussed in the model. At least a sentence or two to discuss this point would be good, e.g. in the Discussion section or somewhere else in the main text.

2) In subsection “Theoretical set-up” the timescale argument is now "correct", but, it would make more sense that the modeller assumes the rates (probability/unit of time) to be of different order from which it follows that the densities operate on different timescales (units of time).

---

## [Author Response]

[Editors’ note: the author responses to the first round of peer review follow.]

Our decision has been reached after consultation between the reviewers. Based on these discussions and the individual reviews below, we regret to inform you that we have decided to reject your paper in its present form. However, we would be willing to consider a revised version of your manuscript that takes into account the extensive reviewer comments.The most important point in such a revision would be to address the criticism that it is unclear in the present version of the manuscript what exactly leads to the evolution of the mating type equilibrium E2. In particular, it is unclear whether the analytical and the simulation results presented in the paper lead to the same conclusions, and how the evolution of the asymmetric state depends on the mutational process. The relationship between the analytical and the simulation results needs to be made much clearer, and both reviewers 1 and 3 suggest doing an invasion analysis to clarify this point, e.g. using adaptive dynamics and pairwise in viability plots.

We have gone through the reviewer’s comments meticulously, have made several revisions and added further analyses to address these concerns. In particular, we have performed a detailed invasion analysis to demonstrate how and when the equilibrium state E2 evolves from E1. We have also computed pairwise invasibility plots and have made an effort throughout the Results section to better illustrate that our analytical and simulated results are in good agreement. Our detailed responses are attached to the reviewer comments below. We believe that the manuscript has greatly improved and appreciate the thorough review and feedback.

Other points you may wish to consider are the evolution of more than two mating types, as well as the possible effects of population-wide ligand and receptor production.

The evolution of the number of mating types is an interesting question. However, we believe that adding work in this direction would divert attention from the core messages of this paper (the evolution of a signaling asymmetry, mating types and suppressed recombination) and falls outside of this scope of the current work. We and others have addressed the evolution of the number of mating types in previous work (Hadjivasiliou and Pomiankowski, 2016 and Constable and Kokko, 2018), and the framework we have developed could be used in future work to investigate the role of ligand-receptor interactions in the evolution of the number of mating types. We discuss this in more detail in the revised manuscript (Discussion section).

The effects of population-wide ligand receptor signals also raise interesting questions. Such effects are to be expected when the ligand is highly diffusible so that its baseline concentration in the background environment is comparable to that produced by individual cells. In this sense our work does not apply to such signals, which are expected to be utilized for sexual chemotaxis. We chose to focus on membrane bound or only locally released (non-diffusible) molecules because this type of signaling is universal amongst isogamous eukaryotes with mating types, whereas diffusible signals are not (see Hadjivasiliou and Pomiankowski, 2016). We justify our choice in the revised manuscript (Introduction).

Reviewer #1:[…]Essential revisions:I think a bit more could be done to demonstrate why the mating type equilibrium (E2) evolves from the monomorphic ancestor (E1), which I think is unclear in a few places (details below). One way to do this would be to take the approach of adaptive dynamics, and assume a monomorphic population (e.g., all individuals make only ligand 1 and receptor 1, v = (1,1,0,0)). You then calculate the fitness of a mutant with trait value v' = (x,y,1-x,1-y) in this population, take the derivatives with respect to x and y, and evaluate at the resident equilibrium (v' = v). A phase plane can then be shown, illustrating the direction of selection acting on small mutations around the resident. [I can send my Mathematica file that produces some examples if the authors wish.] Immediately from this one sees that E1 is locally stable (as the current Figure 2D suggests), as are the other four corners of the phase space. Of particular interest are the basins of attraction for each equilibrium, the effect of parameter values on the size of these basins, and the effects of parameter values on the strength of selection keeping a population within each basin (the "depth" of the basin if you will). The authors have rightly stated that there are multiple basins of attraction (subsection “Evolution of mating types with asymmetric signaling roles”) but in other places claim that E2 evolves from E1 because of strong selection for asymmetry (subsection “Evolution of mating types with asymmetric signaling roles”), implying E1 is instead unstable (i.e., there is no basin of attraction for E1). This idea of instability is implicit in Figure 3D-E, where the idea is that there are threshold values of k^+^, γ, and n that make E1 unstable. I think that the phase plane demonstrates that E1 remains stable for k^+^ finite and γ positive (I've only looked at n=0). Instead of changing the local stability of E1 these parameters change the size of the basin of attraction and its depth, therefore affecting the rate of drift out of E1 (so that "threshold" values of k^+^, γ, and n are just the values that allow this rate to be high enough to see E1 lost by the maximum number of generations simulated). In particular, increasing k^+^ decreases E1's basin of attraction and also makes the basin shallower, greatly facilitating drift out. In contrast, decreasing γ shrinks E1's basin of attraction but makes it deeper, meaning the rate at which E1 transitions to E2 might be less sensitive to changes in γ. This selection surface approach brings a number of new insights (e.g., increasing the size of basin vs. increasing the depth of the basin, the possibility of calculating the amount of drift needed to leave E1), could be used to explain the quantitative patterns in Figure 3D-F (e.g., yellow in Figure 3F is the boundary of the basin of attraction of E1), and highlights that E1 is always a stable state (which is not always clear from the text; subsection “Evolution of mating types with asymmetric signaling roles”). Further investigations along these lines could help illustrate the location and properties of the branching point leading to E2 (e.g., is there are "garden of eden" branching point at v = (0.5, 0.5, 0.5, 0.5)?). Additionally, one might take a quantitative genetic approach and use, for example, diffusion equations to ask what the rate of drift out of E1 is (e.g., Barton and Rouhani, 1987).

Thank you for such an in-depth review and detailed suggestions. We have used an adaptive dynamics approach to clarify the stability of E1. Reviewer 1 is correct in stating that E1 is stable to any single mutation. The population only evolves away from E1 when *two* mutations are present, conferring an asymmetry in ligand production and receptor production in opposite directions. No single mutation can confer an advantage on its own in a population where (*v_L_, v_R_, v_l_, v_r_*) = (1, 1, 0, 0) (Figure 3 and subsection “Dependence of gamete interactions on physical parameters” in revised manuscript). This effect can be thought of as a form of positive epistasis whereby the individual mutations are not favored but confer an advantage when they appear together.

This is discussed in more detail in the revised manuscript (subsection “Dependence of gamete interactions on physical parameters”).

We have made considerable effort to implement the reviewer’s suggestions. Note that this required an analysis considering *two* mutations and we opted to use a combination of analytical and numerical results to address this point. In particular:

1) We show the relative fitness of a mutant in different resident populations using our analytical results. This nicely illustrates that an asymmetry in both the ligand and receptor is required for mating types to evolve.

2) We investigate how the basin of attraction of E1 depends on key parameters (*k^+^*, γ and mutation rates) by considering the effect of small mutations in the ligand and receptor in a population initially at (*v_L_, v_R_, v_l_, v_r_*) = (1, 1, 0, 0) (Figure 5).

3) We also performed an invasion analysis and computed pairwise invasibility plots by investigating how a resident at (*v_L_, v_R_, v_l_, v_r_*)_res_ = (1-dx, 1, dx, 0) is invaded by a mutant at (*v_L_, v_R_, v_l_, v_r_*)_mut_ = (1, 1-dy, 0, dy) (Figure 6A). We find that the population reaches a polymorphic state with the mutant and resident at equal frequencies. We also use this approach to investigate how the population evolves from (*v_L_, v_R_, v_l_, v_r_*) = (0.5, 0.5, 0.5, 0.5) (Figure 6B). Our numerical results are in agreement with our analytical predictions (Figure 3 and Figure 5, Figure 6 are aligned). Note that the resident and mutant in our PIPs confer an asymmetry in opposite directions. This allowed us to produce two dimensional plots that illustrate how an asymmetry in ligand and receptor evolves.

We believe that these modifications and additions to our work have helped clarify our findings. We thank the reviewer for their detailed suggestions and hope that the revisions have adequately addressed the concerns raised above.

Reviewer #2:[…]My only criticism would be that there can be a contradiction between two hypotheses: diffusion of ligands is local, and the use of differential equations with products between molecular concentrations as the rate of contact. This precludes the possibility that ligands produced by the whole population might interact and disturb the signalling between cells. Considering interactions between only two close cells would need more precise modeling of spatialized and stochastic processes. I am not sure that it would change a lot the results, but I can imagine that considering the whole population ligands production could affect a lot of the evolutionary outcomes since it might affect a lot the importance of the self-binding parameter k^+^. It would have been interesting to compare whole population vs. two cells interactions.

We agree that the effects of population-wide ligand receptor signals raise interesting questions. Such effects are to be expected when the ligand is highly diffusible so that its baseline concentration in the background environment is comparable to that produced by individual cells. This is because diffusion gives rise to exponentially decaying profiles that would be unlikely to be comparable in magnitude to the concentration profiles in the vicinity of a single cell. This would not be true if the ligand was very highly diffusible but we restrict this work to consider either ligands that are released locally and effectively do not diffuse, or ligands that remain surface bound (Introduction). In this sense our work does not apply to diffusible signals, which are utilized for sexual chemotaxis or to synchronize entry into meiosis. We chose to focus on membrane bound or only locally released (non-diffusible) molecules because this type of signaling is universal amongst isogamous eukaryotes with mating types, whereas diffusible signals are not (Hadjivasiliou and Pomiankowski, 2016). It is therefore more pertinent to consider membrane bound signals when thinking about the early evolution of mating types. It would be interesting to explore how mating type interactions would be affected in situations where signals are diffusible, and cells have to communicate at a distance (e.g. to coordinate entry into meiosis). We touch on these interesting issues in the manuscript (Discussion section).

Reviewer #3:[…]Essential revisions:1) How does the evolution towards asymmetry kick off?a) It seems that for either of the two initially non-utilized receptors "r" or ligands "l" to have a selective advantage, the other must already be present in the population. Is this true?

This is correct. No single mutation can confer an advantage on its own in a population where (*v_L_, v_R_, v_l_, v_r_*) = (1, 1, 0, 0) (Figure 3 and subsection “Dependence of gamete interactions on physical parameters” in revised manuscript). This effect can be thought of as a form of positive epistasis whereby the individual mutations are not favored but confer an advantage when they appear together. This is discussed in more detail in the revised manuscript (subsection “Evolution of mating types with asymmetric signaling roles”).

b) If true, is it that you can nevertheless get evolution towards asymmetry because the mutation rates are assumed high and so at the mutation-selection balance there are always some "l" and "r" in the population? Would you ever observe evolution towards E2 if mutation rates would be of several orders lower?

Indeed, the mutation rates in our analysis impact whether or not E2 evolves from E1 (Figure 5, Figure 4—figure supplement 1). E2 still evolves when the mutation rates are decreased by an order of magnitude but the basin of attraction for E2 shrinks (Figure 5). Reducing the mutation rate even further is likely to shrink the basin of attraction for E2 further. It follows that larger mutational steps would be required for E2 to evolve when mutation rates are low and populations are small. This is an interesting observation that goes back to the role of genetic drift. We made an effort to clarify these considerations in the revised manuscript (subsection “Evolution of mating types with asymmetric signaling roles”).

c) To gain more insight on the evolution towards asymmetry, can you perform some (analytical) invasion analysis e.g. check the stability of E1 and E2?

This is an important point and we have taken a few steps to further clarify how E2 evolves from E1 (please also see our response to reviewer 1). Because, as the reviewer points out above, more than one mutation is required for the evolution of E2 from E1 we had to use a combination of analytical and numerical methods to study and visualize how E2 evolves from E1. First, we have added a figure that shows the relative fitness of a mutant that appears in different resident populations (Figure 3) based on our analytical results. This was then verified by an invasion analysis using simulation (Figure 5). We show that E1 can be invaded by small mutations that confer signaling asymmetry in opposite directions (Figure 5). We explore how the basin of attraction for E2 changes based on different key parameters. We also used simulation to explore how resident populations with a varying degree of asymmetry are invaded by mutants conferring an asymmetry in the opposite direction to reach a polymorphic equilibrium (Figure 6). We think that these changes have helped clarify and solidify our previous findings and thank the reviewer for their suggestions.

2) It could be helpful to include some simple phase-plane/stability analysis for the model (1-3), at least in the appendix. How do we know the concentration of L and R reach their steady state (4-6)? Also, is this the only steady state?

Figure 2—figure supplement 1 shows the steady state concentration for *L, R* and [*LR*] for varying values of *v_R_* and *v_L_*. We assume that the rate of encounters and the duration of the mating process occur at time scales longer than those of the production and degradation rates of the ligand and receptor. This is a reasonable assumption as it takes many hours for cells to meet and mate whereas degradation rates (determining the timescale for reaching a steady state) are generally faster. There is a unique steady state solution with positive concentrations, this follows from solving the quadradic equations that emerge by setting *d[L]/dt* = *d[R]/dt* = *d[LR]/dt* = 0. We now clarify this in the Materials and methods section.

Also, in (1-3) should it be dt instead of d[R]?

Yes. Very silly typo, now corrected.

3) Subsection “Theoretical set-up”: Should the timescale separation argument be the other way round? Because if cells encounter each other at high rate, they’re within cell concentration of L and R would not have enough time to reach their steady state. Why is this a reasonable assumption?

Reviewer 3 is correct. This was an unfortunate mistake, apologies for the confusion. Now corrected.

4) Is there a reason for not assuming trade-off in v_L_ vs. v_R_? One could imagine that cell can't arbitrarily increase the production of both L and R. Also, wouldn't such a trade-off alleviate the conditions for the evolution of asymmetry?

We assume that there is an upper limit in *v_L_* and *v_R_* (normalized at 1). We did consider imposing a trade-off between *v_L_* and *v_R_*. However, cells require both a ligand and a receptor to be able to mate and so imposing a trade-off between the production of the two seemed somewhat artificial. If we were to impose (*v_L_*+*v_l_*) + (*v_R_*+*v_r_*) < 1, we expect to simply shift the upper limit for *v_L_*+*v_l_* and *v_R_*+*v_r_* to 0.5 (rather than 1) which is why we have refrained from doing so.

5) If you would allow for three different receptors and ligands per cell, would three mating types evolve? Using the same logic, would you get an arbitrary number of mating types if the population would be sufficiently large and the mutation rates sufficiently high?

This is certainly possible. Given our own past work and that of others (e.g. Hadjivasiliou and Pomiankowski, 2016 and Constable and Kokko, 2018), we expect the strength of the interaction between the different receptors and ligands, the mutation rates at which new ligands/receptors appear, the population size and duration of asexual phase to together determine the optimal number of mating types. It would be interesting to explore this further using the current framework but we fear than doing so would divert from the core message of this manuscript. We discuss these ideas in the revised manuscript (Discussion section) and hope to explore these questions within the framework developed here in future work.

[Editors' note: the author responses to the re-review follow.]

The reviewers have discussed the reviews with one another and the Reviewing Editor has drafted this decision to help you prepare a revised submission. The consensus was that this is essentially ready to be accepted for publication, provided that the few points are addressed. The revised submission will not need to be sent out for review again.Reviewer #1:Thanks to the authors for carefully addressing my comments. I'm glad to see some of my ideas implemented, which I think add to the intuition gained here. My remaining comments are relatively minor. I think there is still a bit of care needed in describing the stability of E1 and E2 (both comments in subsection “Evolution of mating types with asymmetric signaling roles”and Discussion section). I also think some care is needed in using "epistasis" to describe frequency dependent selection (subsection “Evolution of mating types with asymmetric signaling roles”), and analogies to fitness valley crossing might be useful.

We have made several edits to address these concerns.

Reviewer #3:In my opinion the authors adequately addressed all the points raised by the reviewers. I recommend this paper for publication, provided the two points below will be taken into account.1) It would be good to mention in the "main" text the population size that was used in the simulations. For example, in Figure 5 where you discuss the different mutation rates, in my opinion the population size should also be present. This leads me to a more general comment that it should be made clearer that the mutation rates used throughout the manuscript are unusually high (or at least I failed to find any mention on this caveat.). My guess is that if they would be decreased below the rates used in the paper, say to 10^-4^ or 10^-6^, the evolution to asymmetry would become extremely difficult, hinting that perhaps something else is going on in the evolution of gamete signalling in addition to what is already discussed in the model. At least a sentence or two to discuss this point would be good, e.g. in the Discussion section or somewhere else in the main text.

This is an important point. In fact, the key reason mutation rates are so important is that the population sizes we consider are relatively small. This introduces drift that leads to the loss of mutants exhibiting an asymmetry before they increase in frequency. We predict that higher population sizes would lead to the evolution of asymmetric signaling for smaller mutation rates. In addition, when the population size is “infinite” the evolution of asymmetry should become independent of the mutation rate, *μ* (or frequency at which the mutant is introduced, *p*). These are interesting effects, but we refrain from studying drift in this work and instead place more emphasis on the relative impact of the physical parameters that dictate signaling interactions.

It is also worth pointing out that unicellular eukaryotes undergo several rounds of asexual growth between each sexual phase (this can vary from tens to thousands of vegetative steps, see Constable and Kokko, 2018 and Hadjivasiliou, Pomiankowski and Kuijper, 2016). The effective mutation rate introducing variation in the production of ligands and receptors between sexual rounds is therefore likely to be several orders of magnitude higher than the actual mutation rate at each vegetative step.

We now explicitly state the relevant population size used (Introduction and in figure legends where relevant) and have added a paragraph in the Discussion section addressing the issues raised above.

2) In subsection “Theoretical set-up” the timescale argument is now "correct", but, it would make more sense that the modeller assumes the rates (probability/unit of time) to be of different order from which it follows that the densities operate on different timescales (units of time).

We have slightly modified the text to reflect this.